# SOLVING PDEs VIA LEARNABLE QUADRATURE

## ABSTRACT

Partial differential Equations (PDEs) are an essential tool across science and engineering. Recent work has shown how contemporary developments in machine learning models can directly help in improving methods for solution discovery of PDEs. This line of work falls under the umbrella of Physics-Informed Machine Learning. A key step in solving a PDE is to determine a set of points in the domain where the current iterate of the PDE's solution will be evaluated. The most prevalent strategy here is to use *Monte Carlo* sampling, but it is widely known to be sub-optimal in lower dimensions. We leverage recent advances in *asymptotic expansions* of quadrature nodes and weights (for weight functions belonging to the modified *Gauss-Jacobi* family) together with suitable adjustments for parameterization towards a data-driven framework for learnable quadrature rules. A direct benefit is a performance improvement in solving PDEs via neural networks, relative to existing alternatives, on a set of problems commonly studied in the literature. Beyond finding a standard solution for an instance of a single PDE, our construction enables learning rules to predict solutions for a given *family* of PDEs via a simple use of hyper-networks, a broadly useful capability.

## 1 INTRODUCTION

Differential equations are an indispensable tool across science, providing a framework for modeling/analyzing diverse physical dynamics. Most real-world settings lead to differential equations where analytical solutions are not possible, but research over decades has led to a mature set of numerical methods which can provide an approximate solution in many cases (Ames, 2014; Trefethen & Bau, 2022). Advances in our understanding of deep neural networks as universal function approximators has led to nice results that span both these topics (Chen et al., 2018; Kidger, 2022). Specifically, a growing body of work in the last five years or so has identified novel architectures, by marrying differential equation solvers with deep learning and these formulations offer surprising new capabilities. For example, one now has access to completely data-driven approaches (Li et al., 2020b;a; Kovachki et al., 2021) which use observational data to estimate the operator for a PDE. For small-sample sizes, we have means of obtaining new class of differential equation solvers that exploit neural networks to encode physical laws (Raissi et al., 2019; Kharazmi et al., 2019).

Roughly speaking, the aforementioned line of work (Karniadakis et al., 2021), discussed in more detail later in §7 can be broadly classified under three main threads: (a) PDE solvers based on neural networks (PINNs) (Raissi et al., 2019); (b) PDE discovery (e.g., symbolic regression) (Holt et al., 2023; d'Ascoli et al., 2023) and (c) operator learning (e.g., Fourier Neural Operator). Li et al. (2020b). This classification is loosely based on the amount of data or physics used to solve/inform the forward/inverse problem, (Boullé & Townsend, 2023). Our paper falls under the first category, where we wish to solve a set (or family) of PDEs which share the same differential operator but vary in other ways. Here, we seek to identify how a learning mechanism can deliver efficiency gains solely based on the shared structure and knowledge of physics; *without the use of any labeled data.*

**PDEs and Quadrature** Consider the following second-order PDE,

$$\frac{\partial^2 u}{\partial x^2} + \frac{\partial^2 u}{\partial y^2} = \log(x)\sin(y) + f(x,y)y^3 \tag{1}$$

One way to find a solution $u$ is to integrate both sides with a test function $v(x,y)$ resulting in an integral equation as follows:

$$\int\int\left(\frac{\partial^2 u}{\partial x^2} + \frac{\partial^2 u}{\partial y^2}\right)v(x,y)\mathrm{d}x\mathrm{d}y = \int\int\left(\log(x)\sin(y) + f(x,y)y^3\right)v(x,y)\mathrm{d}x\mathrm{d}y \tag{2}$$

Such a reformulation enables the use of numerical methods which build-up sums that converge to the integral's true value. This has the added benefit of easing regularity conditions on $u$ and solve the original problem in a weighted sense. A *quadrature method* (Golub & Welsch, 1969) will choose evaluation points (nodes) and corresponding weights to minimize the approximation error. The evaluation points may be constant step/uniform or adaptive: implying either fixed or adaptive *quadrature rules* for estimating the integral.

**Challenges in computing an integral.** In many applications from fluid dynamics (turbulent flow) (Kutz, 2017) to radiation treatment planning (fluence calculation at tissue interfaces) (Lou et al., 2021; Beckham et al., 2002) to materials science (fracture mechanics) (Aliabadi & Rooke, 1991; Rice & Tracey, 1973), the associated data involves irregular behavior including singularities. In continuous monitoring devices, sensors may malfunction. The most rudimentary form of uniformly splitting the domain of integration into equal sub-domains, in many cases, is insufficient owing to the singularity associated with the integrand. Further, even a sophisticated scheme of partitioning, runs into difficulties in the multi-dimensional case. A common solution is to use some variant of Monte Carlo sampling. In higher dimensions, we have no choice but to sample at large and expect the estimated solution to converge to the true solution, given enough runtime. In lower dimensions, Monte Carlo sampling is sub-optimal (Rivera et al., 2022) and several strategies to improve the speed and accuracy of the integral computation are known, the prominent ones being some variant of **adaptive quadrature scheme**, essentially choosing an adaptive grid dependent on the integrand.

**Main Idea.** In this paper, we propose a learnable quadrature scheme utilizing a rich theory based on *orthogonal polynomials* and *asymptotic expansions*. Consider solving a PDE either in its *strong form* or *weak form*. In either case, the end goal is to either (a) determine *which points in the domain* to evaluate the function on (this is the *strong form*) or (b) determine *which test functions to use* (for evaluation of the *weak form*). We can tie these choices to the roots of orthogonal polynomials w.r.t. the modified *Gauss-Jacobi* weight functions. Next, we leverage recent advances in asymptotic expansions of quadrature nodes and weights from (Opsomer & Huybrechs, 2023) to achieve a scheme to compute these *efficiently*.

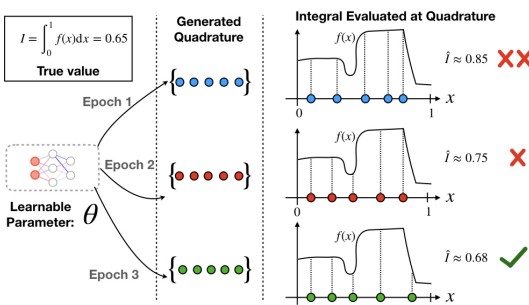

Figure 1: Relevance of learnable quadratures. Given a fixed number of quadrature points, one can setup an optimization problem to update a learnable module based on how good/bad the numerical approximation of the integral is.

**Contributions:** We start with a given PDE and propose a method to learn its solution. The learning is achieved via two separate learnable components. *First* is the actual solution function for the PDE which is parameterized using a neural network. The *second* is a parameterized weight function which in turn induces a family of orthogonal polynomials. Our parameterization of the weight function together with asymptotic expansions when implemented carefully on modern GPUs, can take advantage of parallel compute to generate a very large number *(millions)* of quadrature nodes and weights in constant time. This provides an alternative to Monte Carlo sampling of points for *low-dimensional* problems. We deploy our formulation to solve several commonly used PDEs and achieve better performance that existing adaptive and non-adaptive sampling schemes. Our parameterization of weight function enables learning a quadrature predictor for a family of PDEs (with shared structure), a very useful capability.

## 2 PRELIMINARIES

We briefly review some important concepts that will be useful throughout.

**Orthogonal Polynomials.** Consider a sequence of real-valued polynomials $p_0(x), p_1(x), p_2(x)$ where each $p_n(x)$ is a polynomial of degree $n$. These are *orthogonal* (Olver et al., 2020) with respect to a continuous and non-negative weight function $w(x)$ defined in the interval $(a, b)$ if

$$\langle p_m, p_n \rangle_w = \int_a^b p_m(x) p_n(x) w(x) \mathrm{d}x = \begin{cases} 0 & \text{if } m \neq n, \\ h_n & \text{if } m = n \end{cases} \quad (3)$$

where $h_n$ is a normalization constant. In fact, if $h_n = 1$ for all $n$, then the family is orthonormal.

**Modified Gauss-Jacobi Weight functions.** We use weight functions from the modified Gauss-Jacobi family Opsomer & Huybrechs (2023) to induce our family of orthogonal polynomial (3). These weight functions have the form:

$$w(x) = (1-x)^\alpha (1+x)^\beta h(x); \quad x \in [-1, 1] \tag{4}$$

where $\alpha, \beta > -1$. $h(x)$ is the modifier over the standard Gauss-Jacobi weight function: $w(x) = (1-x)^\alpha(1+x)^\beta$. The only restriction on $h(x)$ is that it should be a strictly positive analytic function.

**Cauchy Residue Theorem.** The Cauchy Residue Theorem (Stein & Shakarchi, 2010) is a powerful tool to compute line integrals of analytic functions over closed curves. Let $f$ be a function that is holomorphic on a simply connected open subset of the complex plane, except possibly at a finite set of points $a_1, \cdots, a_n$(called poles) and $\gamma$ be a positively oriented simple closed curve, then we have:

$$\oint_\gamma f(z)\mathrm{d}z = 2\pi i \sum_{k=1}^n Res(f, a_k); \quad Res(f, a_k) = \frac{1}{2\pi i} \oint_{\gamma_k} f(z)\mathrm{d}z \tag{5}$$

where the quantity $Res(f, a_k)$ is referred to as the *complex residue* of the pole $a_k$. $\gamma_k$ is a positively oriented simple closed curve around the pole $a_k$ *not* including other singularities. While this is the general formula, we will see later that in our specific case, we will only deal with *simple poles*. Then, the formula for the residue simplifies (Stein & Shakarchi, 2010),

$$Res(f, a_k) = \lim_{z \to a_k} (z - a_k)f(z) \tag{6}$$

# 3 STRONG AND WEAK FORMS

We will use $u$ to denote the solution function for a given PDE. Since, $u$ is parameterized/learned, it is commonly called the *trial function*. $\mathcal{L}$ denotes the differential operator acting on $u$. We will mostly deal with non-homogeneous problems and use the function $f$ to denote the non-homogeneity. Furthermore, $g, g_1, \ldots$ will be used to denote functions corresponding to the initial and/or boundary conditions as needed for the PDE at hand. As is standard, we will use $\Omega$ to denote the domain of definition of the PDE and $\partial\Omega$ to denote its boundary.

In its most generic form, an operator $\mathcal{L}$ operating on a function $u$, with a non-homogeneous term $f$ along with boundary and/or initial conditions is

$$\mathcal{L}u = f, \quad \text{in} \quad \Omega; \quad u = g, \quad \text{in} \quad \partial\Omega \tag{7}$$

We used Dirichlet boundary conditions above but these could be specified in terms of derivatives along the normal direction, i.e., Neumann boundary conditions or both. The number of boundary and/or initial conditions needed to completely determine a solution depends on the dimensionality of the PDE in general.

**Solving PDEs.** We examine the canonical form of second order elliptic PDE, the Poisson's equation (in 2 dimensions) as a running example. In 2D-Poisson's equation, the operator $\mathcal{L}$ is the Laplace operator, $\nabla^2$. For $u : \Omega \to \mathbb{R}$, where $\Omega \subset \mathbb{R}^2$ is the domain of interest this is given by:

$$-\nabla^2 u(x,y) = f(x,y), (x,y) \in \Omega; \quad u(x,y) = 0, (x,y) \in \partial\Omega \tag{8}$$

where $f$ is called the *forcing* function. In the homogeneous case where $f = 0$, this becomes the well-known Laplace equation. We will consider $\Omega$ to be the square domain $[-1, 1] \times [-1, 1]$, along with its natural boundary as $\partial\Omega$.

Before presenting our approach, we briefly summarize two approaches to solving the PDE above: via the *strong* and *weak* form of the PDE respectively. This will help emphasize how certain choices (such as **quadrature rule** and **collocation points**) will be key to our parameterization and thereby, learning.

**Strong Form.** A generic PDE shown in (7) is in its **strong form**. Our example in (8) is also in the strong form with a specific choice for the operator. Solving the PDE in its strong form is equivalent to asking that the equations in (8) are satisfied exactly at several points along the domain $\Omega$ and boundary $\partial\Omega$. This can be done by sampling a large number of points distributed uniformly (Monte

Carlo). For simple problems, this approach suffices. But with the inclusion of the non-homogeneous component, a uniform sampling approach can result in poor approximation of the solution, and has been studied extensively (Rivera et al., 2022).

**Weak Form.** While the strong form enforces point-wise exactness, we may only want the property to hold in a "weighted" sense for the entire function. This yields the *weak form*, which involves integration with a *test function*. For the 2D-Poisson equation, using a test function $v(x, y)$ and integrating over the domain, we have

$$\int\int_{\Omega} -\nabla^2 u \quad v \quad \mathrm{d}x\mathrm{d}y = \int\int_{\Omega} f \quad v \quad \mathrm{d}x\mathrm{d}y \tag{9}$$

Solving the PDE in its weak form moves the previous choice of points to a choice of *test functions*. The idea is to use a family of test functions based on the problem at hand and different methods emerge from these choices. These methods are called **Galerkin** methods. One common strategy at this point is to use integration by parts and make use of the boundary conditions to gradually reduce the order of derivatives from the solution $u$ over to the test function $v$, making them symmetric and at the same time easing regularity requirements over the desired solution $u$. If one decides that the test functions in the weak form are Dirac-delta functions, then the Galerkin method reduces to **collocation** and the *weak form* becomes the *strong form* involving differential equations, which then need to be satisfied at the collocation points. In our discussion of the *strong form*, we will refer to the choice of collocation points interchangeably with the choice of test functions.

*Remark* 3.1 (Minimum Principle:). A third route for solving PDEs is the minimization of an **energy** associated with the PDE. This method is often referred to as the **Rayleigh-Ritz** method. But the Galerkin method is general and does not require the problem to be symmetric.

# 4 How to learn Quadrature Rules?

The above discussion helps underscore the importance of choice of *test functions* in the *weak form* or the choice of *collocation points* for solving the PDE in its *strong form*. It is natural to ask: *can the underlying physics play a role in informing these choices?* This section describes such a model by leveraging the rich theory of orthogonal polynomials.

**Learning the weight function:** We consider weight functions to be continuous and positive functions defined in some interval $\mathcal{I}$. Each such weight function $w(x)$ induces a family of orthogonal polynomial (OP) given by (3). Our goal of learning the weight function corresponding to a set of OP is to enable a learnable (or adaptive) quadrature. In order for the method to be practical, we want to compute these efficiently. Our method exploits asymptotic expansions of quadrature nodes for efficiency which are most complete for modified Gauss-Jacobi type weight function. Hence we consider the modified Gauss-Jacobi form (4) and parameterize the modifier $h(x)$ in (4) using a neural network with parameters $\theta$. This keeps the construction simple but offers other interesting benefits we will see shortly. So, our learnable weight function has the form:

$$w_\theta(x) = (1-x)^\alpha(1+x)^\beta h_\theta(x); \quad x \in \mathcal{I} \tag{10}$$

In (10), $\alpha$ and $\beta$ can also be parameterized/learnt but in our experiments, we find that only learning $h_\theta(x)$ suffices. Next, we will see how this learnable weight function nicely ties to the choice of test functions for the weak form and the collocation points in the strong form.

*Remark* 4.1. We will consider the interval of the weight function $\mathcal{I}$ to be the same as $\Omega$, the domain of the PDE. Extensions to higher dimensions can simply be done considering each dimension as independent and stacking the sampled values, this approach is sufficient as we demonstrate in our experiments. More efficient extensions can involve the use of tensor-product or sparse grids (Garcke et al., 2006). The dimensionality of $w_\theta$ will be implicitly determined by the PDE.

**Relation to Solving PDEs in Weak Form:** For the weak form, consider a weighted integral using our weight function $w_\theta$ as the *test function* $v$, on both sides of (9),

$$\int\int_{\Omega} -\nabla^2 u w_\theta \mathrm{d}x\mathrm{d}y = \int\int_{\Omega} f w_\theta \mathrm{d}x\mathrm{d}y \tag{11}$$

To solve the PDE, we must compute both sides of the *integral* efficiently/accurately, specially for weight functions that make maximize deviation from the equality. Therefore, our choice of weight functions crucially helps this computation, as we will describe subsequently.

*Use of Orthogonal Polynomial & Quadrature Rule:* Consider a one-dimensional integral. It is well-known that a $n$-point Gaussian **quadrature rule** can be constructed to yield a very good approximation to the integral of a $2n-1$ degree polynomial, multiplied with the corresponding weight function. For example, if we use the standard (non-modified) Gauss-Jacobi weight function:

$$w(x) = (1-x)^{\alpha}(1+x)^{\beta}; \quad x \in [-1,1] \tag{12}$$

where $\alpha, \beta > -1$. Then, the $n$-th order approximation to the integral is given by:

$$\int_{-1}^{1} f(x)(1-x)^{\alpha}(1+x)^{\beta}\mathrm{d}x \approx \sum_{i=1}^{n} w_i f(x_i) \tag{13}$$

where $x_i$ denotes the $i$-th node and $w_i$ the corresponding weight of the $n$ point Gauss-Jacobi quadrature. Here, $f(x)$ is a smooth function on $[-1,1]$. The nodes $x_i$ are in fact the ***roots*** of the $n$-th degree Jacobi polynomial, which form a family of **orthogonal polynomials** w.r.t. the weight function in (12). We see that our learnable weight function not only provides us a data-dependent choice of *test functions* associated with the *weak form* but also induces a ***learnable quadrature rule*** to compute the integral associated with the *weak form* of the PDE!

**Relation to Solving PDEs in Strong Form:** Recall that solving a PDE in its strong form means enforcing the relationship at several *collocation points* along the domain. A popular method in this category is *orthogonal collocation* (Young, 2019) where the collocation points used are ***roots*** of ***orthogonal polynomials***. Thus, with our choice of *learnable weight function*, $w_\theta$ (which induces a family of orthogonal polynomial by definition (3)) also provides a data-dependent method to sample collocation points to solve a PDE in its strong form!

*Remark* 4.2 (Standard quadrature?)*.* For a family of orthogonal polynomials, the nodes/weights are obtained from a table lookup and is fully data *independent*. On their own, standard quadratures cannot be conveniently utilized for solving a *family* of related PDEs via neural networks.

## 5 LEARNING QUADRATURE RULES EFFICIENTLY

Numerically solving PDEs can be involved (Sewell, 2005), depending on the granularity of discretization used in the algorithm. For the *strong form*, this means efficiently identifying the collocation points and then evaluating the PDE at these points. For the *weak form*, the compute requirement stems from numerically approximating the integral via a quadrature rule. The above discussion laid out a mechanism to learn the weight function. It is not obvious yet whether this can be done efficiently. With our learnable weight functions, (a) in the strong form (7),(8), this means finding the ***roots*** of corresponding orthogonal polynomial which will serve as the collocation points. (b) In the weak form, in using our learnable weight function as the test function, we must determine the quadrature rule and evaluate the integral in the weak form (11) efficiently. Determining the quadrature rule means finding the ***roots*** of an orthogonal polynomial and the corresponding ***quadrature weights***. To do this, we exploit recent advances in ***asymptotic expansions*** of orthogonal polynomials and their roots.

### 5.1 INSTANTIATING ASYMPTOTIC EXPANSIONS

There is a mature literature for fast computation of quadrature nodes and weights corresponding to weight functions of orthogonal polynomials (Townsend, 2015). Over the last few years, this race is dominated by asymptotic expansions (Bogaert, 2014; Townsend et al., 2016). Very recently, (Opsomer & Huybrechs, 2023), proposed *asymptotic expansions* for *generalized* (i.e., modified) versions of canonical weight functions including Gauss-Jacobi and Gauss-Hermite type.

**Division of the Complex Plane.** We briefly present asymptotic expansions of nodes (roots of orthogonal polynomial) and weights of quadrature rule for the modified Jacobi-type weight function (4) based on (Opsomer & Huybrechs, 2023; Opsomer, 2018). The details are not crucial, but useful to appreciate our choice of parameterizations. The reader can check Appendix A and Opsomer & Huybrechs (2023) for more details on the expansions. The derivation of asymptotic expansions starts by dividing the complex plane into four regions, and each region has a different expansion, see Fig.

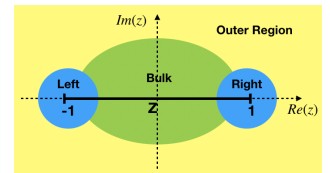

Figure 2: Four different regions of the complex plane for asymptotic expansions.

2. These regions are: the lens covering a bulk of the interval $(-1, 1)$, the two regions on both end-points referred to as left and right disks and everything else is the outer region. Let $\Gamma$ be a shorthand notation for the term $(2n + \alpha + \beta + 1)$ and $n$ refers to the polynomial degree.

*Left endpoint.* The truncated asymptotic expansions of nodes $(x_k)$ and weights $(w_k)$ for hard edge near left endpoint, $x = -1$ are:

$$x_k \sim -1 + \frac{2j_{\beta,k}^2}{(\Gamma + d_0)^2} + \frac{-2j_{\beta,k}^2}{3(\Gamma + d_0)^4}[j_{\beta,k}^2 - 3\alpha^2 - \beta^2 + 1] + \ldots; \quad \frac{w_k}{w(x_k)} \sim \frac{8}{J_{\beta-1}^2(j_{\beta,k})[\Gamma - d_0]^2} + \ldots \tag{14}$$

where $w(x_k)$ is the value of the weight function at node $x_k$, and $c_0$ and $d_0$ are expansion coefficients, described shortly. The expansion uses both (a) the zeros of Bessel functions of order $\beta$ denoted as $j_{\beta,k}$ ($k$-th zero) and the Bessel function of order $(\beta - 1)$ denoted by $J_{\beta-1}$.

*Right endpoint.* For the right end-point, we interchange $\alpha$ and $\beta$ and use $h(-x)$ instead of $h(x)$

*Bulk region.* For expansions in the bulk region, we need to find the leading order term, $t_k$ by solving:

$$\pi \frac{4k + 2\alpha + 3}{4k + 2\alpha + 2\beta + 2} = \arccos(t_k) + \frac{\sqrt{1 - t_k^2}}{\Gamma} \frac{1}{2\pi i} \oint_\gamma \frac{\log(h(\xi))d\xi}{\sqrt{\xi^2 - 1}(\xi - t_k)} \tag{15}$$

Using $t_k$, the truncated asymptotic expansions of nodes and relative weights in the bulk region are

$$x_k \sim t_k + \frac{2\alpha^2 - 2\beta^2 + (2\alpha^2 + 2\beta^2 - 1)t_k}{2[\Gamma + \tau_0]^2} + \ldots; \quad \frac{w_k}{w(x_k)} \sim \frac{\pi\sqrt{1 - t_k^2}}{\Gamma}\Big[2 - \frac{2\tau_1(1 - t_k^2) - 2\tau_0 t_k}{\Gamma}\Big] + \ldots \tag{16}$$

where $\tau_0$, $\tau_1$ are also expansion coefficients.

*Summary.* In (14)–(16), the values of $\alpha, \beta$ correspond to the one used in the modified Gauss-Jacobi weight function (4). The value of $n$ determines the degree of the orthogonal polynomial $p_n(x)$ from the family (3) whose roots we want to compute. Finally, given $n$, $k \in \{1, 2, \ldots, n\}$ corresponds to the $k$-th root of polynomial $p_n(x)$, which is guaranteed to exist and be unique in the interval of definition.

**Expansion Coefficients.** In our description above, we used several coefficients: $c_0, d_0$ and $\tau_0$, and $\tau_1$. While more details are in Appendix A, a synopsis is that the coefficients $c_k, d_k$ stem from series expansion of the modulation function $h(x)$ in (4) (or the parameterized version in (10)) around $z = \pm 1$. The coefficients $\tau_i$ are the series coefficients resulting from the expansion of the contour integral in (15) around the leading order $t_k$ of the $k$-th root of the orthogonal polynomial. The above formulas involve computation of contour integrals, root finding, and series expansions. Computing all these terms exactly within a learnable module will be challenging. We will next perform some simplifications so that the model is amenable to learning.

## 5.2 SIMPLIFICATIONS, ASSUMPTIONS AND IMPLEMENTATION

We now list several assumptions or simplifications needed for an efficient instantiation of the ideas so far. We also present several implementation details.

**Simple Poles.** In order to find the leading order of the $k$-th root, $t_k$, we need to solve a contour integral in (15). We leverage the fact that the roots from the bulk region of interest are real and lie in $(-1, 1)$. Further, we assume that the integrand of the contour integral only has simple poles around $t_k$ so we compute the residue using (6) leading to:

$$\lim_{z \to t_k}(\xi - t_k)\frac{\log(h(\xi))}{\sqrt{1 - \xi^2}(\xi - t_k)} = \frac{\log(h(t_k))}{\sqrt{1 - t_k^2}} \tag{17}$$

Assuming real roots and simple poles, the Cauchy Residue Theorem (5), simplifies (15) as

$$\pi \frac{4k + 2\alpha + 3}{4k + 2\alpha + 2\beta + 2} = \arccos(t_k) + \frac{\log(h(t_k))}{\Gamma} \tag{18}$$

**Root finding and Implicit Function Theorem.** While we avoided computing the contour integral, we need to solve (18) for $t_k$. A solution is available via root finding. We observe that since $t_k$

corresponds to the leading order of the *root* of an orthogonal polynomial, it must exist in $(-1, 1)$. Thus, we can use **bisection method** to find $t_k$ via root finding of the function:

$$F(t_k) = \pi \frac{4k + 2\alpha + 3}{4k + 2\alpha + 2\beta + 2} - \arccos(t_k) - \frac{\log(h(t_k))}{\Gamma} \tag{19}$$

To perform gradient-based updates, we use automatic implicit differentiation from (Blondel et al., 2022), which uses auto-diff of $F(t_k)$ and the implicit function theorem to automatically differentiate through the bisection method.

**How to parameterize?** We parameterize the solution function $u$ as a Multi-Layer Perceptron (MLP). In order to compute the nodes and weights, we need to compute $h(x)$ and the coefficients $c_0, d_0, d_1, \tau_0, \tau_1$, etc. We use a simple MLP for the modulating function $h$ as well as to predict the expansion coefficients for the bulk and edge regions of the nodes and weights.

**Benefits of simplifications/parameterization.** The simplifications above offer multiple benefits. First, we avoid computing contour integrals within a differentiable learning framework. Second, we are able to compute all nodes and weights in **parallel** thereby making the process very *efficient* even for a very large number (*millions*) of nodes. It is worth noting that the exact procedure to compute the nodes (beyond the leading order term) as outlined in Section 4.2 of (Opsomer & Huybrechs, 2023) has a linear time complexity due to the several re-substitutions involved to find the coefficients. Empirically, we verify that the distribution of nodes and weights from the simplifications and parameterization choices does not harm the distribution of quadrature which converges to the expected distribution resembling the roots of polynomial belonging to an orthogonal family.

**Interlacing of roots of orthogonal polynomial.** A naive application of quadrature nodes and weights for the orthogonal polynomial induced by the learnable weight function is insufficient in several cases. This is because with a high degree polynomial (and so, a large number of nodes), within a few epochs, jointly training the solution and quadrature functions, the nodes and weights can overfit to the points being sampled.

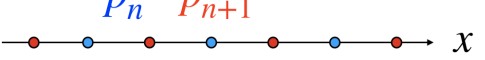

Figure 3: Interlaced red and blue dots on $x$ axis correspond to the roots of polynomial $p_{n+1}(x)$ (degree $n + 1$) and $p_n(x)$ (degree $n$) respectively.

Interestingly, this can be solved via a very useful property of the family of orthogonal polynomials (OP), namely **interlacing** of roots. Given the roots of an OP of degree $n + 1$, the roots of the OP of degree $n$ that belong to the same family are interlaced within the roots of $p_{n+1}$ as shown in Fig. 3. Thus, by utilizing quadrature nodes and weights stemming from varying degree of OP (all of whom correspond to the same weight function being learned), we introduce the desired stochasticity to prevent over-fitting.

**Implementation Details:** To ensure that our learned weight function is positive, we use *softplus* activation on the last layer of the network for $h_\theta(x)$. Further, we found that $\log$ in (19) can lead to vanishing gradients, which is fixed by adding a small amount of noise (order of $e^{-6}$). To avoid invalid quadrature rules due to numerical issues in extreme cases, we used two additional loss components beyond the standard domain and boundary loss terms which are described in the algorithm block below. The loss for enforcing well-behavedness of the learned weight function is given by

$$l_w = \left( \sum_{i=1}^{n} w_i - \int_{-1}^{1} w_\theta(x) \mathrm{d}x \right)^2 + (\sum_{i=1}^{n} w_i - 2)^2 \tag{20}$$

where the first term in (20) enforces the necessary condition that the sum of quadrature weights is equal to the integral of the weight function over the domain of definition. The sec-

---

**Algorithm 1** Training for a single PDE

---

1: **Input:** PDE parameter $\mu$; #epoch: $T$, Learnable models $u_\theta, w_\phi$; PDE Loss $L$ incorporating PDE operator $\mathcal{L}$, inhomogeneous term, initial/boundary condition; regularizer $l_w$.
2: **for** $i = 1$ **to** $i = T$ **do**
3:     Sample, noise: $\zeta \sim \mathcal{N}(0, 1)$
4:     Get $w_\phi$ from $\phi(\zeta)$ or $\phi(\mu)$
5:     Use §5 to get quadrature nodes $\{x_l\}$
6:     Use solution function $u_\theta$ on $\{x_l\}$
7:     Loss:$l = L(u_\theta(x_l)) + l_w(w_\phi)$
8:     Update $u_\theta$ and $w_\phi$ based on $l$
9: **end for**
10: **Output:** Learned models $\theta$ and $\phi$

---

ond term in (20) discourages the quadrature weights from becoming too small. Apart from the above mentioned regularization, we use the standard loss function used in PINN literature (Karniadakis et al., 2021; Cai et al., 2021). Our overall procedure has two trainable components: one is the learnable quadrature module (**LearnQuad**) and the other is the learnable solution function for the given PDE. These can be trained jointly using the loss described above either to simultaneously decrease it

or in a min-max fashion where the quadrature module tries to provide hard to approximate function points in the domain. Empirically, we do not find a large difference in this specific choice of optimization. We provide pseudo-code for training PINN using LearnQuad in Algorithm 1. Our code will be made publicly available.

## 6 EXPERIMENTAL EVALUATIONS

We demonstrate the effectiveness of our proposed framework involving the learnable quadrature module (**LearnQuad**) in solving PDEs next. First we compare the empirical performance of the data adaptive quadrature scheme in solving single PDEs. Thereafter, we describe how the use of hyper-networks can enable **LearnQuad** to efficiently solve a family of PDEs in a data-drive approach.

### 6.1 SOLVING PDES USING LEARNQUAD

**Setup:** We compare the performance of **LearnQuad** in solving several well known PDEs. We benchmark the performance of our proposed data adaptive quadrature scheme against several other adaptive and non-adaptive algorithms (Wu et al., 2023; Lu et al., 2021; Daw et al., 2023). Additional experimental details including the explicit form of the PDEs, hyper-parameter details used in the experiment are included in Appendix B.1. We used the exact same number of points to train the solution model in all methods for a given PDE. Additionally, the solution model in each case had the exact same number of parameters to ensure a fair comparison.

**Result:** We report the $L_2$ relative error (41) as the performance metric following two different experimental settings from Daw et al. (2023) and Wu et al. (2023) over 8 different PDEs in Table 1 and Table 2 respectively. In all but one scenario, **LearnQuad** is able to achieve the best solution function. As can be seen from the numerical results in Table 1 and 2, the $L_2$ relative error for models trained using LearnQuad are better by an order in most cases and also have very little variance (results reported are an average over five runs). We must note that the Diffusion equation used had a very smooth solution and so, almost all methods perform equally well, even with a small number of points. The performance of LearnQuad improves as the number of evaluation points increases, as presented in Table 4; additionally, we observe that LearnQuad can achieve similar performance to other adaptive methods with a much smaller number of points in many cases. All methods have a comparable runtime and memory consumption.

**Summary:** As an adaptive method, **LearnQuad** is highly effective in solving PDEs, leading to performance boost in all cases. The findings reaffirm the usefulness of adaptive methods over non-adaptive ones specifically when the solution function is not well-behaved. **LearnQuad** can be used as drop in replacement for any sampling strategy as a data-driven approach in solving PDEs.

*Remark* 6.1. We use *LearnQuad* in solving a 100 dimensional PDE, a Poisson equation with a very smooth solution (details in Appendix B.2 following Yu et al. (2018)) and achieve a relative $L_2$ error of 0.085 which is similar to using naive Monte Carlo in this setting with relative $L_2$ error of 0.09. This illustrates the viability of *LearnQuad* for high dimensional PDEs. Since the solution is smooth in this particular case, there is no substantial benefit in using a data-driven adaptive method.

*Remark* 6.2. We include additional results on solving PDEs via LearnQuad in their strong form, weak form and also using the energy method in Appendix B.3. These demonstrate that **LearnQuad** is a versatile adaptive scheme which can be used to solve PDEs in multiple reformulations.

| PDE | Convection ($\beta = 30$) | | Convection ($\beta = 50$) | | Allen Cahn |
|---|---|---|---|---|---|
| Epochs. | 100k | 300k | 150k | 300k | 200k |
| PINN (fixed) | $107.5 \pm 10.9\%$ | $107.5 \pm 10.7\%$ | $108.5 \pm 6.38\%$ | $108.7 \pm 6.59\%$ | $69.4 \pm 4.02\%$ |
| PINN (dynamic) | $2.81 \pm 1.45\%$ | $1.35 \pm 0.59\%$ | $24.2 \pm 23.2\%$ | $56.9 \pm 9.08\%$ | $0.77 \pm 0.06\%$ |
| Curr Reg (Krishnapriyan et al. (2021)) | $63.2 \pm 9.89\%$ | $2.65 \pm 1.44\%$ | $48.9 \pm 7.44\%$ | $31.5 \pm 16.6\%$ | – |
| CPINN (fixed) (Wang et al. (2022)) | $138.8 \pm 11.0\%$ | $138.8 \pm 11.0\%$ | $106.5 \pm 10.5\%$ | $106.5 \pm 10.5\%$ | $48.7 \pm 19.6\%$ |
| CPINN (dynamic) (Wang et al. (2022)) | $52.2 \pm 43.6\%$ | $23.8 \pm 45.1\%$ | $79.0 \pm 5.11\%$ | $73.2 \pm 3.6\%$ | $1.5 \pm 0.75\%$ |
| RAR-G (Lu et al. (2021)) | $10.5 \pm 5.67\%$ | $2.66 \pm 1.41\%$ | $65.7 \pm 1.77\%$ | $43.1 \pm 28.9\%$ | $25.1 \pm 23.2\%$ |
| RAD (Nabian et al. (2021)) | $3.35 \pm 2.02\%$ | $1.85 \pm 1.90\%$ | $66.0 \pm 1.55\%$ | $64.1 \pm 11.9\%$ | $0.78 \pm 0.05\%$ |
| RAR-D (Wu et al. (2023)) | $67.1 \pm 4.28\%$ | $32.0 \pm 25.8\%$ | $82.9 \pm 5.96\%$ | $75.3 \pm 9.58\%$ | $51.6 \pm 0.41\%$ |
| $L^\infty$ | $66.6 \pm 2.35\%$ | $41.2 \pm 27.9\%$ | $76.6 \pm 1.04\%$ | $75.8 \pm 1.01\%$ | $1.65 \pm 1.36\%$ |
| R3 (Daw et al. (2023)) | $1.51 \pm 0.26\%$ | $0.78 \pm 0.18\%$ | $1.98 \pm 0.72\%$ | $2.28 \pm 0.76\%$ | $\mathbf{0.83 \pm 0.15}\%$ |
| Causal R3 ( Daw et al. (2023)) | $2.12 \pm 0.67\%$ | $0.75 \pm 0.12\%$ | $5.99 \pm 5.25\%$ | $2.28 \pm 0.76\%$ | $0.71 \pm 0.007\%$ |
| LearnQuad | $\mathbf{0.78 \pm 0.002}\%$ | $\mathbf{0.68 \pm 0.02}\%$ | $\mathbf{0.79 \pm 0.02}\%$ | $\mathbf{0.76 \pm 0.01}\%$ | $0.87 \pm 0.01\%$ |

Table 1: $L_2$ relative error over benchmark PDEs with using 1000 collocation points. **LearnQuad** achieves best accuracy in 4 out of 5 settings.

| PDE
No. of points | | Diffusion
30 | Burgers'
2000 | Allen-Cahn
1000 | Wave
2000 |
|---|---|---|---|---|---|
| Non-adaptive | Grid | $0.004 \pm 0.001$ | $0.12 \pm 0.04$ | $0.88 \pm 0.06$ | $0.42 \pm 0.09$ |
| | Random | $0.005 \pm 0.002$ | $0.13 \pm 0.03$ | $0.32 \pm 0.14$ | $0.48 \pm 0.07$ |
| | LHS | $0.003 \pm 0.002$ | $0.18 \pm 0.15$ | $0.32 \pm 0.04$ | $0.61 \pm 0.13$ |
| | Halton | $0.002 \pm 0.0006$ | $0.06 \pm 0.02$ | $0.18 \pm 0.05$ | $0.46 \pm 0.06$ |
| | Hammersley | $0.001 \pm 0.0007$ | $0.07 \pm 0.05$ | $0.17 \pm 0.05$ | $0.31 \pm 0.09$ |
| | Sobol | $0.002 \pm 0.002$ | $0.08 \pm 0.03$ | $0.20 \pm 0.10$ | $0.49 \pm 0.09$ |
| Adaptive | Random-R | $0.12 \pm 0.06$ | $1.69 \pm 1.67$ | $0.55 \pm 0.34$ | $0.72 \pm 0.90$ |
| | RAR-G (Lu et al. (2021)) | $0.0009 \pm 0.0008$ | $0.12 \pm 0.04$ | $0.53 \pm 0.19$ | $0.81 \pm 0.11$ |
| | RAD (Nabian et al. (2021)) | $0.0019 \pm 0.00097$ | $0.02 \pm 0.00$ | $0.08 \pm 0.06$ | $0.09 \pm 0.04$ |
| | RAR-D (Wu et al. (2023)) | $0.004 \pm 0.0041$ | $0.03 \pm 0.01$ | $0.09 \pm 0.03$ | $0.29 \pm 0.04$ |
| | **LearnQuad** | $\mathbf{0.0005 \pm 0.0001}$ | $\mathbf{0.003 \pm 0.002}$ | $\mathbf{0.03 \pm 0.008}$ | $\mathbf{0.005 \pm 0.0006}$ |

Table 2: $L_2$ relative error (mean $\pm$ standard deviation) of the trained solution function obtained while using different adaptive and non-adaptive methods. The lowest error for each problem is denoted in boldface. Model trained via **LearnQuad** achieves the lowest $L^2$ relative error in all case.

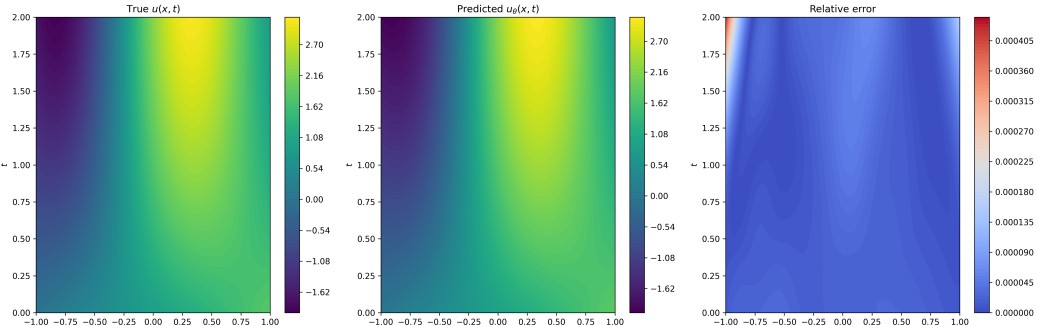

Figure 4: PDEs from family of wave equation: (left) numerical solution, (center) solution predicted using **LearnQuad** and (right) relative error between them.

## 6.2 SOLVING A FAMILY OF PDEs VIA LEARNQUAD

**Setup:** Given the effectiveness of **LearnQuad** in solving a given PDE, we now utilize our framework to tackle a harder problem. We consider a *family* of PDEs, where our end goal is to solve a PDE given a particular choice of forcing function and/or PDE hyper-parameters and initial and/or boundary conditions. Based on (7), a family of PDEs corresponding to differential operator $\mathcal{L}$ refers to the set of triplets, $\{(f_i, g_i, u_i)\}_{i=1}^N$, where each $i$-th PDE satisfies:

$$\mathcal{L}u_i = f_i, \quad \text{in} \quad \Omega; \quad u_i = g_i, \quad \text{in} \quad \partial\Omega \tag{21}$$

We only assume access to $f_i, g_i$'s and $\mathcal{L}$. We note that $f_i$ denotes the forcing function and/or PDE hyper-parameter and $g_i$ denotes the initial and/or boundary condition corresponding to the $i$-th PDE which is governed by operator $\mathcal{L}$. As an example, if $f$ has the following parametric form:

$$f_\kappa(x) = -(a(\pi\theta)^2 \sin(\pi\theta x) + b(\pi\psi)^2 \cos(\pi\psi x)) \tag{22}$$

where $\kappa = \{a, b, \theta, \phi\} \sim p$; we can sample $\kappa \sim p$ to obtain $f_i$'s (similarly for $g_i$'s) and then learn to solve for PDEs corresponding to $\mathcal{L}$.

Our full training pipeline is shown in Fig. 5. We use two hyper-networks with learnable parameter(s) $\theta$ and $\phi$ which provide parameters of the weight function $w_\kappa(x)$ and solution function $u_\kappa(x)$ respectively based on the input $\kappa \sim p$. This weight function is then used to generate a suitable quadrature $\{x_l\}_\kappa$ for the PDE corresponding to $\kappa$. These are then used to evaluate the $\mathcal{L}u_\kappa(x)$ and $f_\kappa(x)$ and minimize the loss based on the strong form in (21). The pseudo-code for using **LearnQuad** in solving a family of PDE is given in the Appendix B.4. We demonstrate the effectiveness of *LearnQuad* in this setting via several different PDEs: Laplace, Advection, Burger's, Wave and Heat equation. Experimental details for each case is included in Appendix B.4.

**Result:** The above scheme incorporating **LearnQuad** to train on a family of PDEs achieves excellent generalization performance and is faster to converge than one using Monte Carlo sampling. We report the absolute relative error compared to the numerical solution obtained using the same number of domain points used for *LearnQuad* in Table 3.

| PDE | Wave | | Advection | | Heat | | Burgers' | |
|---|---|---|---|---|---|---|---|---|
| I.C./B.C. | Eqn.63 | Eqn.64 | Eqn.67 | Eqn.68 | Eqn.58 | Eqn.59 | Eqn.73 | Eqn.74 |
| Test Error | 9.9e-6 | 3.3e-5 | 1.9e-5 | 7.9e-5 | 2.1e-4 | 3.5e-4 | 2.8e-4 | 3.3e-4 |

Table 3: Absolute Relative error on the test set on four different PDEs. For each PDE we present results on two different families corresponding to the different Initial Conditions (I.C.) and/or Boundary Conditions (B.C.) as mentioned. More details are in Appendix B.4.

As can be seen, the model is able to generalize very well on the test set PDEs in each case. We visualize the solution function and corresponding error for the family corresponding to wave equation and viscous Burgers' equation in Figure 4. Additional details of PDE used and more visualizations are included in Appendix B.4.

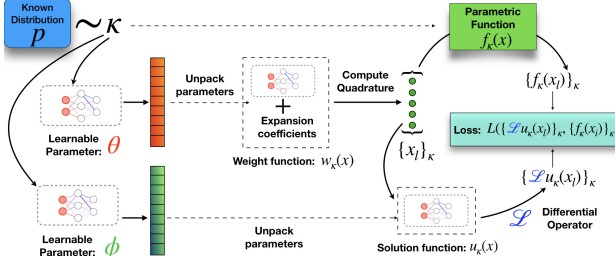

**Summary:** Learning quadratures to solve a family of PDEs is extremely beneficial. Once we have trained our model, given any new forcing function and boundary/initial condition, we **avoid** the need to train a separate

Figure 5: Framework for solving a family of PDEs governed by operator $\mathcal{L}$, where forcing function/external condition/PDE hyper-parameter are parameterized by known distribution $p$. Two hyper-networks with parameters $\theta$ and $\phi$ generate the weight and solution function.

model. We can generate the solution in a single forward pass. This is *extremely efficient* in terms of time and completely removes the need to store and process different solution functions separately.

*Remark* 6.3 (Distinction with Operator Learning:). While the end result of learning for a family of PDEs may appear similar to operator learning, the problem settings are actually very different. Operator learning uses paired data $(f_i, u_i)$ and is a supervised learning framework where the operator is learned. On the other hand, our method using *LearnQuad* is completely unsupervised in the sense that we *only* have access to $f_i$'s,$g_i$'s and complete knowledge of the shared operator, $\mathcal{L}$.

## 7 RELATED WORK

Beyond the literature described in §1, a large body of work focuses on discovering solutions to PDEs using neural networks. We mention some ideas and how our framework is different. In contrast to Physics Informed Neural Networks (PINN)s Raissi et al. (2019), we use data-dependent sampling of collocation points. While Variational-PINN (Kharazmi et al., 2019) and hp-VPINN (Kharazmi et al., 2021) solve PDEs in weak form, they use a careful choice of test functions, which is learnable in our case. Finally, compared to Deep-Ritz (Yu et al., 2018), our method does not need the minimum energy principle to be applicable. In fact, as noted earlier, we provide a novel way to learn solving PDEs which is complementary to existing works. We also acknowledge recent ideas focused on or adjacent to adaptive quadrature (Rivera et al., 2022; Omella & Pardo, 2024; Lau et al., 2024), which either directly try to optimize node locations thereby resulting in a much larger optimization problem or fall-back to problem-specific regularizer(s) which may limit their applicability.

## 8 CONCLUSIONS

We present a data-driven approach to solve PDEs, by exploiting new results of fast quadrature computation using asymptotic expansions and recent capabilities of implicit function differentiation. We demonstrate the incorporating our learnable quadrature scheme, **LearnQuad** while solving a PDE can lead to performance improvement over exisiting adaptive and non-adaptive sampling schemes across a diverse set of PDEs. Additionally, we show that incorporation of **LearnQuad** is extremely beneficial when solving a family of PDEs – where the alternative would be to deploy a Monte Carlo based scheme for each instance individually. Our proposed hyper-network based approach generates the solution to a PDE instance from a given family in just a single forward pass.

While our proposed framework is independent of the dimensionality of the problem, incorporation of techniques such as sparse grids, can potentially yield better performance by exploiting the structure better. It would be interesting to combine learnable quadratures with quasi-Monte Carlo technique for potential benefits.

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

## A    ASYMPTOTIC EXPANSION

In this section, we list the full expansion of nodes and weights used for experiments in the paper:

For the left hard edge at $x = -1$

$$x_k \sim -1 + \frac{2j_{\beta,k}^2}{(\Gamma + d_0)^2} + \frac{-2j_{\beta,k}^2}{3(\Gamma + d_0)^4}[j_{\beta,k}^2 - 3\alpha^2 - \beta^2 + 1] + \dots$$

$$+ \frac{-j_{\beta,k}^2}{6(2n + \alpha + \beta + 1 + d_0)^5}[16(d_0 - 3d_1)j_{\beta,k}^4 + 3(4\alpha^2 - 1)c_0 + (12\alpha^2 + 8\beta^2 - 5)d_0$$

$$- 6(4\beta^2 - 1)d_1] + \dots + O(n^{-8})$$

$$\frac{w_k}{w(x_k)} \sim \frac{8}{J_{\beta-1}^2(j_{\beta,k})[\Gamma - d_0]^2}$$

$$+ \frac{8}{3J_{\beta-1}^2(j_{\beta,k})[2n + \alpha + \beta + 1 - d_0]^4}[3\alpha^2 + \beta^2 - 1 - 2j_{\beta,k}^2]$$

$$- \frac{2[32(d_0 - 3d_1)j_{\beta,k}^2 + 3(4\alpha^2 - 1)c_0 + (12\alpha^2 + 8\beta^2 - 5)d_0 - 6(4\beta^2 - 1)d_1]}{3J_{\beta-1}^2(j_{\beta,k})[2n + \alpha + \beta + 1 - d_0]^5} + \dots + O(n^{-8}) \tag{23}$$

where $\Gamma = 2n + \alpha + \beta + 1$

For the bulk region:

$$x_k \sim t_k + \frac{2\alpha^2 - 2\beta^2 + (2\alpha^2 + 2\beta^2 - 1)t_k}{2[\Gamma + \tau_0]^2} - \frac{1}{4[2n + \alpha + \beta + 1 + \tau_0]^3}\left(4(\alpha^2 - 1)c_0 + \right.$$

$$+ 4(\beta^2 - 1)d_0 + 8(\alpha^2 - \beta^2)\tau_0 - 4(\alpha^2 - \beta^2)\tau_1 + 2(2\alpha^2 + 2\beta^2 - 1)\tau_1 t_k^3$$

$$+ 2[(2\alpha^2 + 2\beta^2 - 1)\tau_0 + 2(\alpha^2 - \beta^2)\tau_1]t_k^2$$

$$+ [4(\alpha^2 - 1)c_0 - 4(\beta^2 - 1)d_0 + 4(3\alpha^2 + \beta^2 - 1)\tau_0 - 2(2\alpha^2 + 2\beta^2 - 1)\tau_1]t_k\right) + h.o.t. \tag{24}$$

$$\frac{w_k}{w(x_k)} \sim \frac{\pi\sqrt{1 - t_k^2}}{\Gamma}\left[2 - \frac{2\tau_1(1 - t_k^2) - 2\tau_0 t_k}{\Gamma}\right] + \frac{1}{(2n + \alpha + \beta + 1)^2}\left(2\tau_1^2 t_k^4 + 4\tau_0\tau_1 t_k^3\right.$$

$$\left. - 4\tau_0\tau_1 t_k + 2(\tau_0^2 - 2\tau_1^2)t_k^2 + 2\alpha^2 + 2\beta^2 + 2\tau_1^2 - 1) + h.o.t.\right]$$

The coefficients $c_k$ and $d_k$ are given by:

$$c_k = \frac{1}{2\pi i}\oint_\gamma \frac{\log(h(\xi))}{(\xi^2 - 1)^{1/2}}\frac{d\xi}{(\xi - 1)^{k+1}} \tag{25}$$

$$d_k = \frac{1}{2\pi i}\oint_\gamma \frac{\log(h(\xi))}{(\xi^2 - 1)^{1/2}}\frac{d\xi}{(\xi + 1)^{k+1}} \tag{26}$$

Hereafter, we refer the reader to (Opsomer, 2018; Opsomer & Huybrechs, 2023) for further detail on the asymptotic expansions pertinent to modified Gauss-Jacobi weight functions.

## B EXPERIMENT DETAILS

### B.1 SOLVING PDEs VIA LEARNQUAD

Below we describe the four PDEs used in the experimental results of Table 2.

#### B.1.1 DIFFUSION EQUATION

We consider the following one dimensional diffusion equation:

$$\frac{\partial u}{\partial t} = \frac{\partial^2 u}{\partial x^2} + e^{-t}\left(-\sin(\pi x) + \pi^2 \sin(\pi x)\right), \quad x \in [-1, 1],\ t \in [0, 1], \tag{27}$$

$$u(x, 0) = \sin(\pi x), \tag{28}$$
$$u(-1, t) = u(1, t) = 0, \tag{29}$$

with domain $[-1, 1]$ in space and $[0, 1]$ in time. The exact solution to this diffusion equation is given by $u(x, t) = \sin(\pi x)e^{-t}$, which is a smooth one and hence all methods as illustrated in Table 2 perform reasonably well. The model used in this case is a fully connected neural network with hidden layers of width 32 and depth 3.

#### B.1.2 BURGER'S EQUATION

We consider the following Burger's equation:

$$\frac{\partial u}{\partial t} + u\frac{\partial u}{\partial x} = \nu\frac{\partial^2 u}{\partial x^2}, \quad x \in [-1, 1],\ t \in [0, 1], \tag{30}$$

$$u(x, 0) = -\sin(\pi x), \tag{31}$$
$$u(-1, t) = u(1, t) = 0, \tag{32}$$

where $\nu$ is the viscosity of the fluid and $u$ is the desired flow velocity. In our experiments, we have used $\nu = 0.01/\pi$ which results in a non-smooth solution. The model used in this case is a fully connected neural network with hidden layers of width 64 and depth 3.

#### B.1.3 ALLEN-CAHN EQUATION

The Allen-Cahn PDE considered in our experiments is as follows:

$$\frac{\partial u}{\partial t} = D\frac{\partial^2 u}{\partial x^2} + 5(u - u^3), \quad x \in [-1, 1],\ t \in [0, 1], \tag{33}$$

$$u(x, 0) = x^2 \cos(\pi x), \tag{34}$$
$$u(-1, t) = u(1, t) = -1, \tag{35}$$

We use a value of $D = 0.001$ as the diffusion coefficient in the PDE. The model used in this case is a fully connected neural network with hidden layers of width 64 and depth 3.

#### B.1.4 WAVE EQUATION

We consider the following one dimensional wave equation:

$$\frac{\partial u}{\partial t} = c^2\frac{\partial^2 u}{\partial x^2}, \quad x \in [0, 1],\ t \in [0, 1], \tag{36}$$
$$u(0, t) = u(1, t) = 0, \quad t \in [0, 1] \tag{37}$$

$$u(x, 0) = \sin(\pi x) + \frac{1}{2}\sin(4\pi x), \quad x \in [0, 1] \tag{38}$$

$$\frac{\partial u}{\partial t}(x, 0) = 0, \quad x \in [0, 1] \tag{39}$$

with $c = 2$, where $c$ is the velocity of the wave. The solution in this specific choice demonstrates a multi-scale behavior in both space and time dimension and is as follows:

$$u(x, t) = \sin(\pi x)\cos(2\pi t) + \frac{1}{2}\sin(4\pi x)\cos(8\pi t) \tag{40}$$

The model used in this case is a fully connected neural network with hidden layers of width 100 and depth 5.

### B.1.5 OTHER DETAILS

The number of parameters used for the learnable weight function in the LearnQuad module was roughly 500 parameters in all cases. All neural networks were implemented using fully connected layers with $\tanh$ as the activation function. All experiments were performed on a single NVIDIA 2080 Ti GPU. The number of epochs used for diffusion PDE was $100k$ while for Burger's, Wave and Allen-Cahn PDE they were run for $200k$ epochs. This was determined empirically based on convergence of the $L_2$ relative error. We used a learning rate of 1e-3. As noted in Algorithm 1, one could either use a noise sampled from the standard normal or the PDE specific parameters as an input to the learnable quadrature module and results are not too different, but slightly better on using standard normal noise as input. We find jointly optimizing both the LearnQuad and solution model provides very good performance without the need for a sophisticated min-max optimization scheme. The $L_2$ relative error reported in the paper is computed as the following:

$$L_{2error} = \frac{||u_\theta - u||_2}{||u||_2} \tag{41}$$

Here, $u_\theta$ is the learned solution function and $u$ is the "ground truth" solution. In a small number of cases where the true solution is available in a closed form we use that as $u$ or we use $u$ to be a numerical solution achieved using a traditional numerical scheme (finite difference). In any case, the test error is evaluated on a uniform grid of a much higher density $(10x)$ than the number of points used in the training scenario. We emphasize that the "ground truth" solution is not used in any form during the training period.

### B.1.6 PERFORMANCE OF LEARNQUAD

We enumerate the performance of **LearnQuad** with increasing number of points in three different PDEs, (outlined previously) in Table 4. As expected, the performance in terms of $L^2$ relative error improves on increasing the number of points. Note that the solution to the diffusion equation is very smooth and hence even a very small number of points can lead to very good performance.

| Diffusion Equation | | Allen-Cahn Equation | | Wave Equation | |
|---|---|---|---|---|---|
| No. of Points | $L^2$ Error | No. of Points | $L^2$ Error | No. of Points | $L^2$ Error |
| 20 | 0.0013 | | | 200 | 0.017 |
| 25 | 0.0007 | 200 | 0.0444 | 500 | 0.0076 |
| 30 | 0.0004 | 700 | 0.0331 | 1500 | 0.0064 |
| 35 | 0.0003 | | | 2500 | 0.0052 |
| 40 | 0.0002 | 1500 | 0.0280 | 3500 | 0.0044 |

Table 4: $L^2$ Relative Error for Different PDEs with varying number of points used by LearnQuad. Performance improves on increasing the number of points as expected.

### B.1.7 PERFORMANCE OF LEARNQUAD WITH VARYING HYPER-PARAMETER

We investigate the performance of LearnQuad with varying the hyper-parameters of $\alpha$ and $\beta$ in the modified Gauss-Jacobi weight function from (equation 10). We report the test performance in terms of the relative $L^2$ relative error in Table 5. We observe minor variations in the performance of LearnQuad based on the choice of these hyper-parameters.

| $(\alpha, \beta)$ | Diffusion | Wave | Convection |
|---|---|---|---|
| $(2, 2)$ | 0.0004 | 0.0058 | 0.7299 |
| $(3, 3)$ | 0.0005 | 0.0056 | 0.7163 |
| $(1, 2)$ | 0.0004 | 0.0065 | 0.7207 |
| $(2, 1)$ | 0.0006 | 0.0044 | 0.7323 |
| $(10, 10)$ | 0.0007 | 0.0062 | 0.6774 |

Table 5: $L^2$ Relative Error for Different PDEs with varying $\alpha$ and $\beta$ in the modified Gauss-Jacobi weight function used by LearnQuad. We observe there are minor variations based on the choice of these hyper-parameters.

## B.2 SOLVING HIGH DIMENSIONAL PDE

We consider the following high-dimensional Poisson equation.

$$-\Delta u = -200, \quad x \in (0, 1)^{100} \tag{42}$$

$$u(x) = \sum_{i=1}^{100} x_i^2, \quad x \in \partial(0, 1)^{100} \tag{43}$$

which is in a 100 dimensional space with the true solution being $u(x) = \sum_{i=1}^{100} x_i^2$. As mentioned in Section 6.1 this is very smooth solution and hence both adaptive and non-adaptive methods perform equally well. This experiment, demonstrates that LearnQuad is not restricted to low dimensional problems. We used a fully connected neural network with hidden layers having a depth of 3 and width of 100 as the solution model with tanh as the activation function. We used 1000 points in 100 dimensions. For this problem, our training took 18 seconds to converge in 300 epochs. After this, evaluating the trained model on any given resolution takes 0.0065 seconds. The test errors were computed with respect to the true analytical solution which is readily available in this case.

## B.3 SOLVING PDES IN STRONG, WEAK AND ENERGY FROM VIA LEARNQUAD

We describe empirical evaluations of our proposed framework using **LearnQuad**. We show results for solving a single given PDE via all three main approaches: (a) the strong form, (b) weak form and (c) minimum principle.

**(A) Numerical experiments with Strong Form.** We begin by deploying our learnable quadrature first in solving PDEs via the strong from described in §3. We consider two operators: (a) 1D-Laplace and (b) $\frac{d^2}{dx^2} + \frac{d}{dx}$. For each of these operators, we consider two different non-homogeneous conditions. As shown in Fig. 6, the results of the predicted and true solution function $u$ coincide exactly in all four cases. Both the domain and boundary loss are of the order of e-5, the same as the baseline (PINN). (Raissi et al., 2019).

For the 1D-Laplace operator, we use the following two functions as the non-homogeneous terms:

$$f(x) = 2 - \sin(x) + 60x - 2((\cos(x))^2 - (\sin(x))^2) \tag{44}$$

$$f(x) = 90(x^8) - (4\pi^2)\sin(2\pi x) - (4\pi^2)\cos(2\pi x) \tag{45}$$

For the 1D operator $\frac{d^2}{dx^2} + \frac{d}{dx}$, we use the following two functions:

$$f(x) = 3x^2 + 2\pi x \cos(\pi x^2) + \frac{1}{2} + 6x + 2\pi \cos(\pi x^2) - (4\pi x^2)\sin(\pi x^2) \tag{46}$$

$$f(x) = 3x^2 + 6\pi x \cos(3\pi x^2) + \frac{1}{2} + 6x + 6\pi \cos(3\pi x^2) - 36\pi^2 x^2 \sin(3\pi x^2) \tag{47}$$

*Remark* B.1. Using Monte Carlo based sampling to solve PDEs (as in PINNs) can have undesirable outcomes when dealing with irregular boundary, hence adaptive quadrature methods have been proposed very recently (Omella & Pardo, 2024). Our method is data-driven does not suffer from such challenges since the quadratures are adaptive by design.

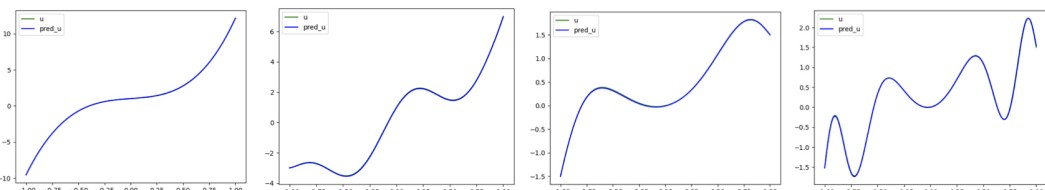

Figure 6: True/Predicted solution functions. 2 right-most two plots for 2 different conditions on the 1D-Laplace operator. 2 left-most two plots for solutions to 2 settings for the operator $\frac{d^2}{dx^2} + \frac{d}{dx}$

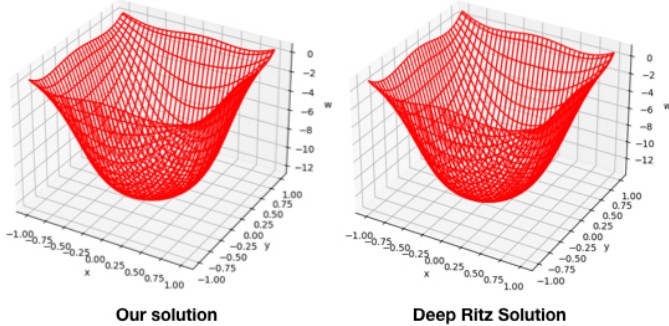

Our solution          Deep Ritz Solution

Figure 7: Comparison of solution curves obtained via the proposed learnable method and baseline method of Deep Ritz. Both methods perform equally well.

**(B) Numerical experiments with Weak Form.** We next apply our learnable quadratures to solve PDEs written in their weak form as described in §3. Here, we consider the following two 1-D operators: (a) 1D-Laplace and (b) $\frac{d^2}{dx^2} - \frac{d}{dx}$. Similar to the strong form, we present results for two different conditions for each operator. Again, we see from Fig. 8 that, the predicted and true solution function in all cases coincide almost exactly. In terms of the domain and boundary loss, these are of the same order of $e^{-3}$ as the baseline method of hp-VPINN (Kharazmi et al., 2021).

Since the weight functions can be global, in using them as test functions to solve the weak form, we can end up with a global test function. Avoiding this is possible via several schemes: one could either choose a multitude of such test functions or simply use sub-domain splitting as suggested in hp-VPINN over VPINN Kharazmi et al. (2019). Due to its simplicity, we choose the latter in our experiments.

For the 1D-Laplace operator, we use the following two functions as the non-homogeneous terms:

$$f(x) = 2 - \sin(x) + 60x - 2(\cos^2(x) - \sin^2(x)) \tag{48}$$

$$f(x) = 90(x^8) - 4\pi^2 \sin(2\pi x) - 4 * \pi^2 \cos(2\pi x) \tag{49}$$

For the 1D operator $\frac{d^2}{dx^2} - \frac{d}{dx}$, we use the following two functions:

$$f(x) = 6x + 2\pi \cos(\pi x^2) - 4\pi x^2 \sin(\pi x^2) - (3x^2 + 2\pi x \cos(\pi x^2) + \frac{1}{2}) \tag{50}$$

$$f(x) = 6x + 6\pi \cos(3\pi x^2) - 36\pi^2 x^2 \sin(3\pi x^2) - (3x^2 + 6\pi x \cos(3\pi x^2) + \frac{1}{2}) \tag{51}$$

*Remark* B.2. For solving PDEs in their strong and weak forms as presented above, we adopt a two stage training scheme. In the first stage, the asymptotic quadrature is learned and in the second stage these learned quadratures are used to either provide orthogonal collocation points in the strong form or test function(s) for the weak form. Our overall procedure is otherwise unchanged.

**(C) Energy Method.** We now demonstrate the utility of learnable quadrature for solving a PDE where the loss function is derived based on the minimum energy principle.

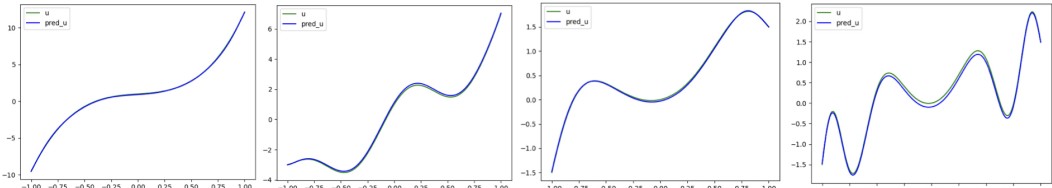

Figure 8: True/Predicted solution functions. 2 right-most two plots for 2 different conditions on the 1D-Laplace operator. 2 left-most two plots for solutions to 2 settings for the operator $\frac{\mathrm{d}^2}{\mathrm{d}x^2} - \frac{\mathrm{d}}{\mathrm{d}x}$

We consider the 2D-Laplace equation: $\Delta u = -100$ with zero boundary conditions on a square domain: $[-1, 1] \times [-1, 1]$. In the energy form, the loss function has the form

$$L(u) = \int\int_{\Omega} \left( \frac{1}{2}|\Delta u|^2 - fu \right) \mathrm{d}x\mathrm{d}y + \beta \int\int_{\partial\Omega} u^2 \mathrm{d}x\mathrm{d}y \tag{52}$$

where $\beta$ is a penalty term on the second component denoting the boundary loss. The first component is the loss on the domain. We use our learnable quadrature to approximate both integrals in equation 52 and compare the solution obtained with the baseline method of Deep-Ritz (Yu et al., 2018) with same number of parameters, running each for roughly 400 epochs. As can be seen from Fig. 7, our method achieves comparable performance, with approximate loss value $-2000$ in both case.

*Remark* B.3. Since our proposed method is, in essence, a data-driven way to sample points, it shows its utility in solving PDEs via *all three* formulations as demonstrated above, where the basic framework remains the same. In the *strong* form, it provides **orthogonal collocation** points. In the *weak form*, it provides **test functions** (which induce the quadrature rules). Finally, in the *energy form* it is used to directly provide a **quadrature rule**.

### B.4 FAMILY OF PDE VIA LEARNQUAD

We specify the details of the family of PDEs which were solved using **LearnQuad** and the procedure outlined in 6.2. The overall algorithm is presented in Algorithm 2.

In all experiments, we used 500 parameters each for the hyper-networks predicting the weight function and solution function as outlined in Section 6.2. Specifically, we used a MLP-based neural network with depth 5; width 100 and $\tanh$ as the activation function. The number of parameters to encode the actual solution function were kept smaller than 20. Using a learning rate of 0.0001, in all cases, the methods took less than 10k epochs to converge. For each family, we sampled 100 instances of the PDE and used a train/test split of $80/20$. We used 600 points as a standard number of points to sample from **LearnQuad**.

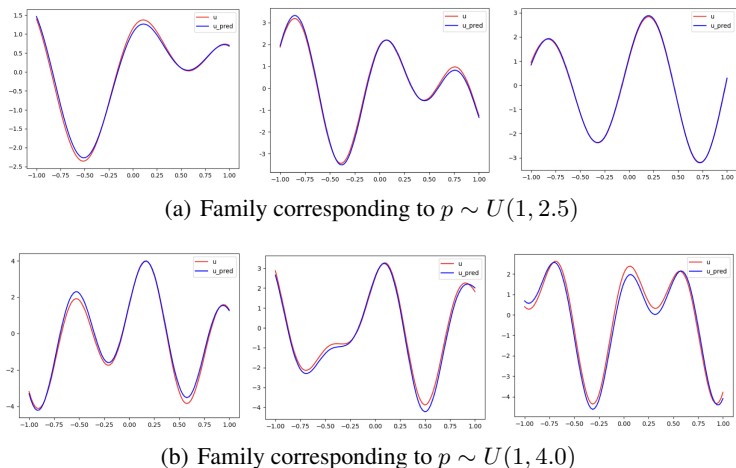

(a) Family corresponding to $p \sim U(1, 2.5)$

(b) Family corresponding to $p \sim U(1, 4.0)$

Figure 9: Test results for solving PDEs from two different family each with 1-D Laplace operator. The second family 9(b) has more variance than the first. In all cases, the predicted solution almost coincides with the original solution

---

**Algorithm 2** Training for a family of PDEs

---

1: **Input:** Operator $\mathcal{L}$; distribution $p$; parametric form of forcing function $F$ and boundary/initial conditions $B$. #epochs: $T$, Training size: $n$, Learnable modules $\theta, \phi$; PDE Loss function $L$ and regularization loss $l_w$ equation 20
2: **Compute:** Generate Training Set, $S = \{\}$
3: **for** $i = 1$ **to** $i = n$ **do**
4:     Sample $\kappa \sim p$
5:     Get $f_\kappa$ from $F$, Get $b_\kappa$ from $B$
6:     $S$.append($\kappa, f_\kappa, b_\kappa$)
7: **end for**
8: **Compute:** Training Loop
9: **for** $i = 1$ **to** $i = T$ **do**
10:     **for** each $(\kappa, f_\kappa, b_\kappa) \in S$ **do**
11:         Get $w_\kappa$ and $\{\tau_0, \tau_1, c_0, d_0, d_1\}_\kappa$ from $\theta(\kappa)$
12:         Get $u_\kappa$ from $\phi(\kappa)$
13:         Use §5 to get quadrature nodes $\{x_l\}_\kappa$
14:         Loss:$l = L(\{\mathcal{L}u_\kappa(x_l)\}_\kappa, \{f_\kappa(x_i)\}_\kappa) + l_w$
15:         Gradient based update for $\theta$ and $\phi$ based on $l$
16:     **end for**
17: **end for**
18: **Output:** Learned modules $\theta$ and $\phi$

---

### B.4.1 FAMILY OF LAPLACE EQUATION

Here, we consider the 1D-Laplace operator which has the following parametric representation for the non-homogeneous function:

$$f_\kappa(x) = -(a\pi^2\mu^2 \sin(\pi\mu x) + b\pi^2\nu^2 \cos(\pi\nu x)) \tag{53}$$

where, $\kappa = \{a, b, \mu, \nu\}$ belong to different distribution. In our experiments, we choose these distributions as uniform, but our method can handle any distribution. In Figure 9 we show the performance on the test set. It can seen that the predicted solution is very close to the true solution.

### B.4.2 FAMILY OF HEAT EQUATION

We consider the one dimensional heat equation and sample the heat diffusivity, $c$; initial distribution, $f$; and two boundary conditions, $T_l$ and $T_r$. The PDE along with initial and Dirichlet boundary

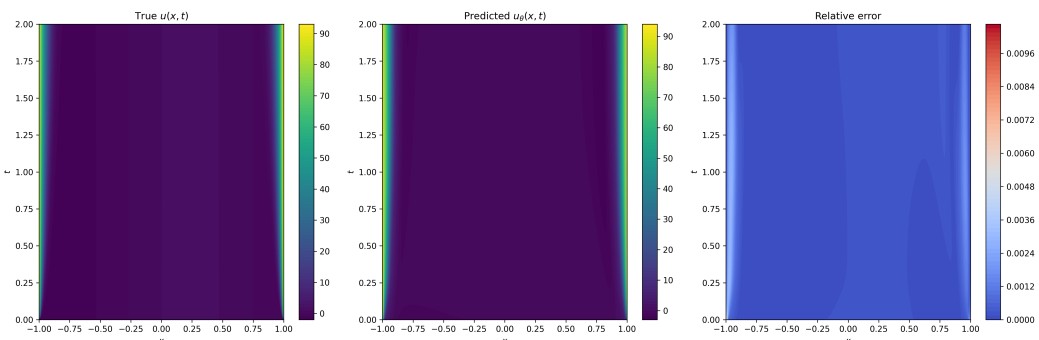

Figure 10: (Left)True solution, (Center) Predicted solution and (Right) Relative error for an instance from the test set of the family of heat equation using (54)-(57) and (59)

conditions is given as follows:

$$\frac{\partial u}{\partial t} = c^2 \frac{\partial^2 u}{\partial x^2}, \quad x \in [-1, 1],\ t \in [0, 2] \tag{54}$$

$$u(-1, t) = T_l, \quad t \in [0, 2] \tag{55}$$

$$u(1, t) = T_r, \quad t \in [0, 2] \tag{56}$$

$$u(x, 0) = f(x), \quad x \in [-1, 1] \tag{57}$$

We perform experiments, with two choices for the initial distribution:

$$f(x) = mx + n \tag{58}$$

$$f(x) = a \sin(\pi \theta x) + b \cos(\pi \phi x) \tag{59}$$

We present a visualization of the true (numerical) solution obtained using the same number of domain points as LearnQuad, the predicted solution and their relative error in Figure 10.

### B.4.3 FAMILY OF WAVE EQUATION

We consider the 1D wave equation and sample the wave speed, $c$ and the initial position, $f$ and velocity, $g$. The PDE along with initial conditions is given below:

$$\frac{\partial^2 u}{\partial t^2} = c^2 \frac{\partial^2 u}{\partial x^2}, \quad x \in [-1, 1],\ t \in [0, 2] \tag{60}$$

$$u(x, 0) = f(x), \quad x \in [-1, 1] \tag{61}$$

$$\frac{\partial u}{\partial t}(x, 0) = g(x), \quad x \in [-1, 1] \tag{62}$$

We perform experiments, with the following two sets of initial conditions:

$$f(x) = mx, \quad g(x) = a + x; \quad x \in [-1, 1] \tag{63}$$

$$f(x) = mx + n, \quad g(x) = a \sin(\pi \theta x) + b \cos(\pi \phi x); \quad x \in [-1, 1] \tag{64}$$

We present visualization of the true (numerical) solution obtained using the same number of domain points as LearnQuad, the predicted solution and their relative error in Figure 11 and Figure 12.

### B.4.4 FAMILY OF ADVECTION EQUATION

We consider the one dimensional advection equation and sample the advection speed, $c$ and the initial position $f$. The PDE along with the initial conditions is given by:

$$\frac{\partial u}{\partial t} + c \frac{\partial u}{\partial x} = 0, \quad x \in [-1, 1],\ t \in [0, 2] \tag{65}$$

$$u(x, 0) = f(x), \quad x \in [-1, 1] \tag{66}$$

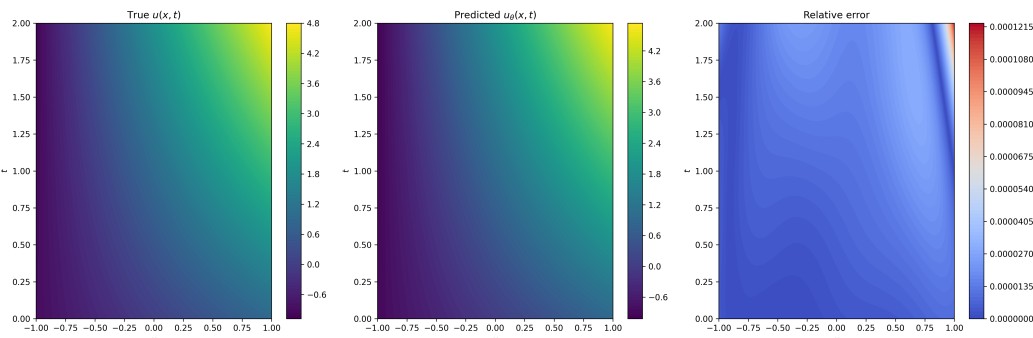

Figure 11: (Left)True solution, (Center) Predicted solution and (Right) Relative error for an instance from the test set of the family of wave equation using (60)-(62) and (63)

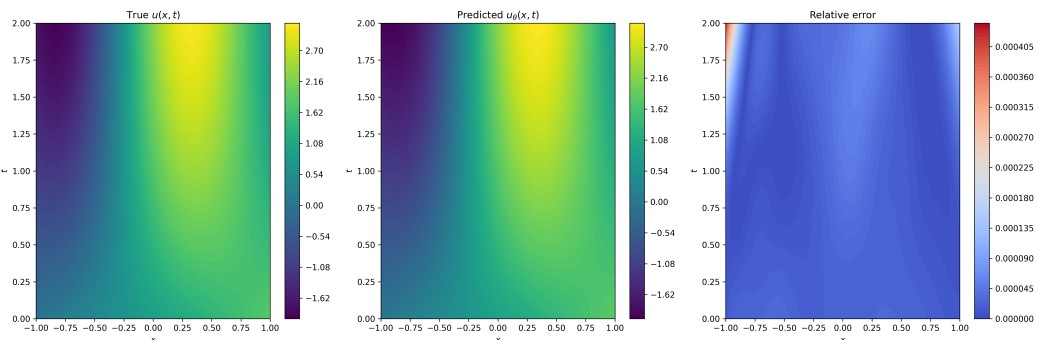

Figure 12: (Left)True solution, (Center) Predicted solution and (Right) Relative error for an instance from the test set of the family of wave equation using (60)-(62) and (64)

We conduct experiments with the two following choices for the initial displacement:

$$f(x) = mx + n, \quad x \in [-1, 1] \tag{67}$$
$$f(x) = a\sin(\pi\theta x) + b\cos(\pi\phi x), \quad x \in [-1, 1] \tag{68}$$

We present visualization of the true (numerical) solution obtained using the same number of domain points as LearnQuad, the predicted solution and their relative error in Figure 13 and Figure 14.

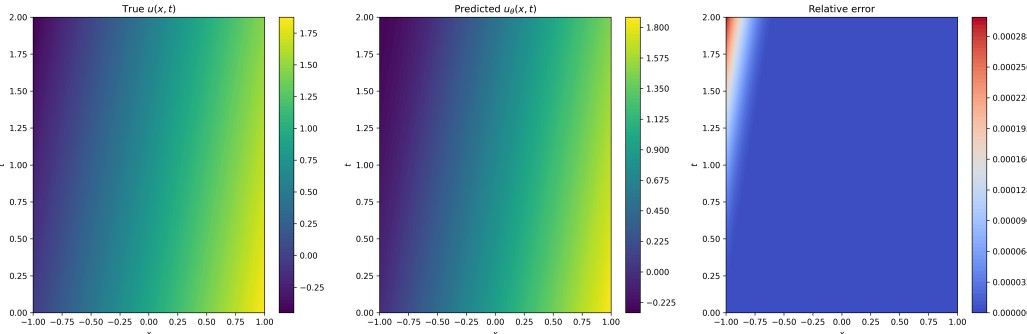

Figure 13: (Left)True solution, (Center) Predicted solution and (Right) Relative error for an instance from the test set of the family of advection equation using (65)-(66) and (67)

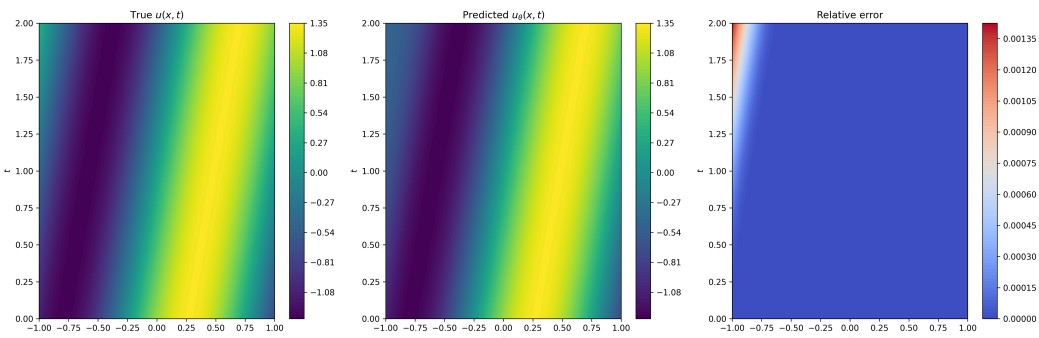

Figure 14: (Left)True solution, (Center) Predicted solution and (Right) Relative error for an instance from the test set of the family of advection equation using (65)-(66) and (68)

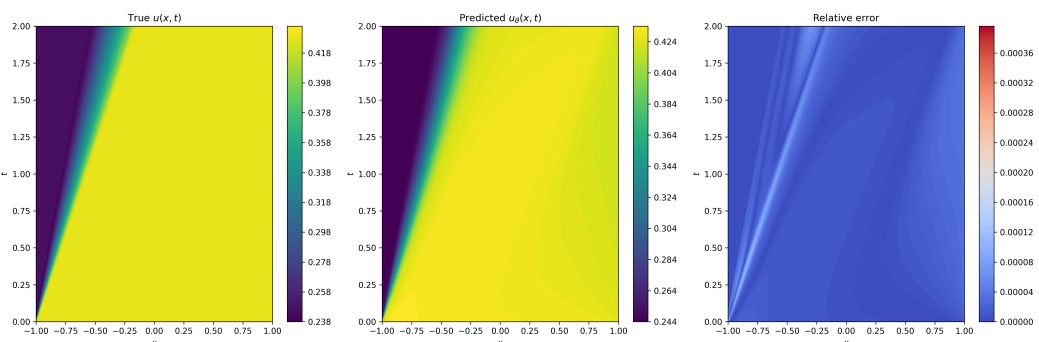

Figure 15: (Left)True solution, (Center) Predicted solution and (Right) Relative error for an instance from the test set of the family of viscous Burgers' advection equation using (69)-(72) and (73)

### B.4.5 FAMILY OF BURGER'S EQUATION

We consider the one dimensional viscous Burgers' equation which is a non-linear PDE. We sample the diffusivity coefficient, $c$; initial velocity distribution, $f$; and the two boundary conditions, $T_l$ and $T_r$. The PDE along with the initial and boundary conditions is given by:

$$\frac{\partial u}{\partial t} + u \frac{\partial u}{\partial x} = c^2 \frac{\partial^u}{\partial x^2}, \quad x \in [-1, 1],\ t \in [0, 2] \tag{69}$$

$$u(-1, t) = T_l, \quad t \in [0, 2] \tag{70}$$

$$u(1, t) = T_r, \quad t \in [0, 2] \tag{71}$$

$$u(x, 0) = f(x), \quad x \in [-1, 1] \tag{72}$$

We consider the following two different choices for the initial condition:

$$f(x) = m, \quad x \in [-1, 1] \tag{73}$$

$$f(x) = a \exp^{-bx^2}, \quad x \in [-1, 1] \tag{74}$$

We present visualization of the true (numerical) solution obtained using the same number of domain points as LearnQuad, the predicted solution and their relative error in Figure 15, Figure 16 and Figure 17.

## C ADDITIONAL DISCUSSIONS

### C.1 PINNS AND CLASSICAL SOLVERS

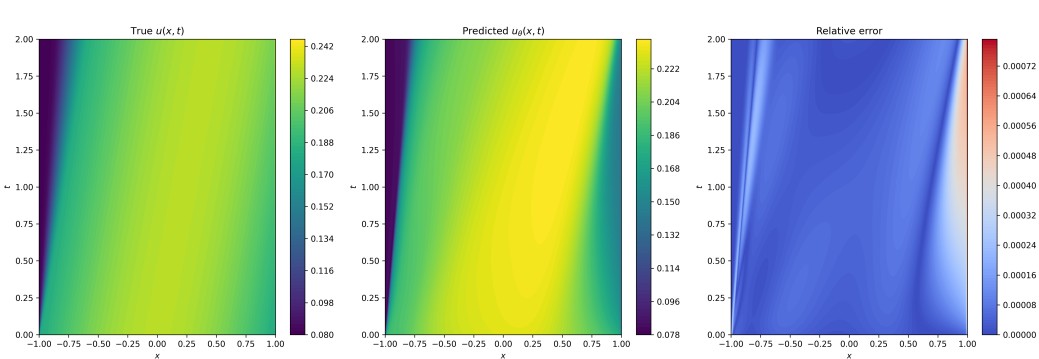

Figure 16: (Left)True solution, (Center) Predicted solution and (Right) Relative error for an instance from the test set of the family of viscous Burgers' equation using (69)-(72) and (74)

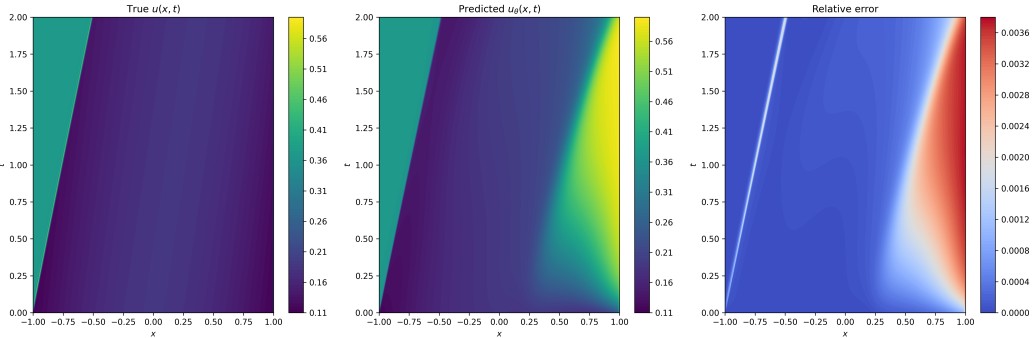

Figure 17: (Left)True solution, (Center) Predicted solution and (Right) Relative error for an instance from the test set of the family of viscous Burgers' equation using (69)-(72) and (74)

Our proposed method is not designed to compete with classical solvers. While for simple problems, classical methods are indeed effective, one motivation behind the sizable PINN literature is an alternative which is advantageous in many scenarios: (a) PINN based solutions are mesh-independent; (b) they rely on automatic-differentiation which are easier to implement; and (c) can handle non-linearity effectively given the universal function approximation properties of neural networks. We note that PINN based methods never use "classical solution" as the ground truth in the training procedure at all. It is only used to evaluate a test time performance metric. This is needed in cases where the PDE solution is not given in a closed form, which is true for most scenarios.

## C.2   PINN LOSS OVER PDE SOLVER

Our object of interest in this work is PINN. PINNs provide a mesh independent solution, are more amenable to non-linearities and are easier to scale and implement. Hence, PINNs offer many benefits in several cases and for this reason, are being studied extensively. Next, we justify our choice of PINN loss instead of a PDE solver.

While the learnable quadrature rule is amenable for classical solvers, there are several issues. Suppose we use a classical solver instead of a PINN loss. This means that for each update of parameters $\theta$ in leanrable weight function $w_\theta$ (which induces the quadrature) we will need to (i) generate quadrature points using current $w_\theta$, (ii) solve the system of equations (either implicitly using a solver or iteratively), (iii) compute some loss/quality and (iv) update $\theta$ to improve this metric. This poses several challenges. Explicit (iterative) solvers are memory-intensive when unrolling across time steps, sensitive to numerical instabilities, thereby requiring fine time steps and increased computational cost. When differentiating through a numerically unstable solver, the gradients can become inaccurate or blow up. Implicit solvers demand solving linear or nonlinear systems. Computing Jacobians for implicit differentiation requires significant computational resources. Furthermore, matrix inversion or solving linear systems as part of implicit differentiation introduces high computational overhead. Hence, we can agree that integrating PDE solvers into neural network modules presents challenges for both explicit and implicit solvers due to the above mentioned issues in computing gradients which are necessary to update the models via back-propagation. Therefore, in order to make learnable quadrature feasible – the main goal of this work – we leverage the PINN loss which is more suited for the end-to-end learning framework.

Another aspect worth mentioning is regarding the setup for a family of PDEs. Without a scheme to learn the common structure shared between different instances of the PDE, it would require solving each instance separately at each desired resolution. To conclude, the choice of the loss is important not for solving individual PDEs, but for permitting the learning of quadrature rules that can then be used across multiple problems/solution schemes.

## C.3   CONTRAST WITH OTHER ADAPTIVE METHODS FOR PINNS

Our main contribution is not just solving PDEs, but learning how to optimally sample points based on the PDE's structure. We emphasize that advantages stem directly from our core theoretical contribution: the learnable weight function that induces problem-specific quadrature rules. This is fundamentally different from both classical adaptive methods and other existing ML approaches like R3Daw et al. (2023), RARLu et al. (2021), RADWu et al. (2023); all of whom invariably rely on computing error estimates through residual-based estimators or gradient thresh-holding which are problem-specific, need to be chosen carefully, and sometimes may need to solve additional local problems. Instead we adaptively learn where refinement may be needed in an end-to-end fashion in conjunction with the PINN loss and no additional explicit error estimation is required.

