# OpenReview forum: "Solving PDEs via learnable quadrature"
_ICLR.cc/2025/Conference — Submitted to ICLR 2025_

### Official Review · Reviewer_jfde · 2024-11-02

**Soundness:** 3
**Presentation:** 3
**Contribution:** 3
**Rating:** 6
**Confidence:** 4

**Summary:**

This article is a combination of numerical analysis and machine learning. Based on the classic Gauss-Jacobi iteration method, it uses data-driven (machine learning) methods to set relevant weights and node information, making numerical simulation (numerical integration and numerical solutions of differential equations) more accurate and efficient.

**Strengths:**

Advantages:

1. Combining traditional numerical analysis methods with modern machine learning is a very novel point. It is not difficult to see from the article that the new framework not only includes the advantages of traditional iterative methods, but also improves the shortcomings of traditional iterative methods to a certain extent. (Especially in residual and weak form)

2. The experimental results are very extensive, and the experimental examples are very extensive. (Appendix content)

3. Greedy optimization in details improves performance (reflected in saving optimization/training time)

**Weaknesses:**

Disadvantages:

1. I want to know if the convergence theory of traditional methods (including convergence, stability and consistency) is still valid under your model if the weights and interpolation points are learned (I think you can add more mathematical proofs), and whether the machine learning method can improve the iteration step size.

2. In terms of experiments, since various learning solvers are emerging, you can compare the results with some classic machine learning PDE solvers on the market, such as PINN (although I know you mentioned this, but I think you can compare widely), DeepONet, kernel method, etc.

3. When mentioning the weights of the classic numerical analysis method, I want to know what the distribution of sampling (you also mentioned MCMC in the article) should be like if data-driven means (machine learning) are used, that is, how to ensure a relatively balanced optimization in different dimensions (whether high or low dimensions)? Or is there any potential mathematical explanation for this?

**Questions:**

The experimental results are very good and general, and I hope authors could add/do more theoretical analysis. I will be happy to improve my score in subsequent discussions.

**Details Of Ethics Concerns:**

None. This is original work and there are no ethical issues

---

> ### Author Response · Authors · 2024-11-22
>
> Dear Reviewer jfde,
>
> We thank you for the detailed feedback. We appreciate your overall positive view of our work. We address your concerns below:
>
> Q 1) *I want to know if the convergence theory of traditional methods (including convergence, stability and consistency) is still valid under your model if the weights and interpolation points are learned (I think you can add more mathematical proofs), and whether the machine learning method can improve the iteration step size.*
>
> Regarding convergence, stability and consistency, our procedure can sample points more effectively and as such this does not affect the theory. For example, the weight function learns to concentrate quadrature points around rapid change or singularities. This is analogous to how one would choose from existing weight functions corresponding to classical orthogonal polynomials; our learned function simply does this adaptively in a data-driven way. But establishing complete convergence theory analogous to classical methods needs additional work. While our empirical results demonstrate nice convergence behavior, proving specific rates of convergence would need specializing to a narrow class of PDEs, and analyzing how the learned weight function affects or improves approximation properties under various assumptions. This would be interesting as a follow-up work but may be of limited interest to the machine learning community.
>
> Q 2) *In terms of experiments, since various learning solvers are emerging, you can compare the results with some classic machine learning PDE solvers on the market, such as PINN (although I know you mentioned this, but I think you can compare widely), DeepONet, kernel method, etc.*
>
> Good point. We should note that our method complements PINNs by providing a data-dependent way to sample points for learning the PINN based solution. This has lead to improvement in performance which we have demonstrated in Section 6. For comparison with the operator learning framework (DeepONet, etc.), the reviewer will agree that our method is completely unsupervised, whereas operator learning requires paired data (for supervised learning) which can sometimes be difficult to obtain and is expensive. We have mentioned this distinction in Remark 6.3.
>
> Based on reviewer 5qE3’s suggestion, we compared our method with a more recent baseline and observed that LearnQuad achieves better performance than every other adaptive and non-adaptive method on 4 out of 5 new PDEs. We report the numerical results below and will include this in the final version.
>
> | **PDE**          | **Convection (β=30)** | **Convection (β=30)** | **Convection (β=50)** | **Convection (β=50)** | **Allen-Cahn**   |
> |--------------------|-----------------------|-----------------------|-----------------------|-----------------------|------------------|
> | **Epochs**  |     100k                 | 300k                 | 150k                 | 300k                 | 200k            |
> | PINN Fixed                   | 107.5 ± 10.9         | 107.5 ± 10.7         | 108.5 ± 6.38         | 108.7 ± 6.59         | 69.4 ± 4.02      |
> | PINN Dynamic                  | 2.81 ± 1.45          | 1.35 ± 0.59          | 24.2 ± 23.2          | 56.9 ± 9.08          | 0.77 ± 0.06      |
> | Curr Reg                     | 63.2 ± 9.89          | 2.65 ± 1.44          | 48.9 ± 7.44          | 31.5 ± 16.6          | --               |
> | CPINN Fixed                   | 138.8 ± 11.0         | 138.8 ± 11.0         | 106.5 ± 10.5         | 106.5 ± 10.5         | 48.7 ± 19.6      |
> | CPINN Dynamic                | 52.2 ± 43.6          | 23.8 ± 45.1          | 79.0 ± 5.11          | 73.2 ± 3.6           | 1.5 ± 0.75       |
> | RAR-G                        | 10.5 ± 5.67          | 2.66 ± 1.41          | 65.7 ± 1.77          | 43.1 ± 28.9          | 25.1 ± 23.2      |
> | RAD                          | 3.35 ± 2.02          | 1.85 ± 1.90          | 66.0 ± 1.55          | 64.1 ± 11.9          | 0.78 ± 0.05      |
> | RAR-D                        | 67.1 ± 4.28          | 32.0 ± 25.8          | 82.9 ± 5.96          | 75.3 ± 9.58          | 51.6 ± 0.41      |
> | $L^\infty$                  | 66.6 ± 2.35          | 41.2 ± 27.9          | 76.6 ± 1.04          | 75.8 ± 1.01          | 1.65 ± 1.36      |
> | R3                            | 1.51 ± 0.26          | 0.78 ± 0.18          | 1.98 ± 0.72          | 2.28 ± 0.76          | **0.83 ± 0.15**  |
> | Causal R3                    | 2.12 ± 0.67          | 0.75 ± 0.12          | 5.99 ± 5.25          | 2.28 ± 0.76          | **0.71 ± 0.007** |
> | LearnQuad                    | **0.78 ± 0.002**     | **0.68 ± 0.02**      | **0.79 ± 0.02**      | **0.76 ± 0.01**      | 0.87 ± 0.01      |

---

> ### Author Response · Authors · 2024-11-22
>
> Q 3) *When mentioning the weights of the classic numerical analysis method, I want to know what the distribution of sampling (you also mentioned MCMC in the article) should be like if data-driven means (machine learning) are used, that is, how to ensure a relatively balanced optimization in different dimensions (whether high or low dimensions)? Or is there any potential mathematical explanation for this?*
>
> Thank you for bringing this up, it is a very good point. For MCMC, the distribution of sampling for all dimensions is uniform. On the other hand, for our proposed method, the points are sampled from the distribution induced by the learned weight function via its family of orthogonal polynomials. This is beneficial because each potential dimension can be treated either independently (different weight functions) or the same based on whether the problem is symmetric along these dimensions. Hence, it allows us to leverage more knowledge about the PDE when learning its solution. The fact that the weight functions (if chosen differently for different dimensions) can be optimized jointly helps us to ensure that each dimension is treated equally. We are happy to include these discussions in the final version of the paper.
>
> We sincerely hope that we have been able to address your concerns. We are very happy to provide further clarification at any point.

---

> ### Author Response · Authors · 2024-11-28
>
> Dear Reviewer jfde,
>
> We really appreciate the time you've invested in our paper. If there is anything else we can answer or clarify, please let us know. Many thanks.

---

> > ### Comment · Reviewer_jfde · 2024-12-01
> >
> > Your answer has cleared my doubts to a certain extent. It seems that the LearnQuad does have broader advantages. I am very grateful and I have improved my score.

---

> ### Author Response · Authors · 2024-12-02
>
> Dear Reviewer jfde,
>
> We thank you for reading our response and engaging with us. We are grateful for your willingness to improve the score. We will appreciate if you can update the recommendation/rating.

---

### Official Review · Reviewer_64fQ · 2024-11-02

**Soundness:** 2
**Presentation:** 1
**Contribution:** 2
**Rating:** 3
**Confidence:** 3

**Summary:**

This work aims to improve sampling of neural network to solve partial differential equations using physics-informed neural networks. Instead of using Monte-Carlo sampling, which is benefitial in high-dimensions but sub-optimal in low-dimensions, the authors propose a data-driven methods for learning quadrature rules. This relies on a family of orthogonal polynomials of the form $w(x) = (1-x)^\alpha (1+x)^\beta h(x)$, where $h$ is a positive function, and quadrature nodes associated with the Gauss-Jacobi weight functions. They apply their method on a family of PDE and show that they can reach lower error than alternative sampling method.

**Strengths:**

The introduction is well-written and the proposed method displays good performance over the alternative sampling techniques on all the problems considered.

**Weaknesses:**

I find the paper very confusing and a large time is spent in reviewing background material (e.g. on orthogonal polynomials). It is not clear why one cannot simply use a standard Gaussian quadrature (if the solution is smooth) or a deep Galerking method with a variational formulation based on finite elements (if the solution is not smooth) and why one needs to learn the weight function.

The experiments seem promising but it is not clear how the L^2 error is computed: is it using the learnt weight function or on a uniform grid. In any case, this requires some convergence analysis to analyze the convergence rate of the proposed quadrature rule as the number of sampling point increases. Otherwise, I do not know why I should use this over any standard quadrature rules.

There seems to be quite a lot of confusion in the presentation between weak form and strong form (see the questions).

**Questions:**

- Line 49: I don't understand this paragraph: integrating the equation would just give you an equation between two real number, the first one would correspond to the average of u over the domain. After integration, it is impossible to recover u. To solve (1), one can either solve it in the strong form, or in the weak form after integrating by a test function (smooth with compact support).
- The choice of the quadrature rule is basically equivalent to the choice of basis functions to approximate u. If one uses a finite element method, then one can employ a quadrature rule on each mesh element that integrates polynomial exactly up to the FEM degree, hence making zero integration error, and similarly for spectral methods.
- line 98: I would refrain from making such novelty statement
- I do not understand why introducing Cauchy residue theorem is necessary or Section 3, and 5.1.
- line 154: this requires the function spaces for u and f. If $f=0$, then the solution is trivial as $u=0$ with the choice of boundary conditions.
- Is there a connection with adaptive mesh refinement?
- line 203: the choice of this particular weight function is not motivated
- line 215: fix formatting for the equation
- line 244: "enable learning", I do not understand what this means
- line 248: a general statement like this is meaningless as certain equations are very easy to solve. Training a neural network to solve a PDE is often (always?) more computational intensive.
- How do you compute the L2 error in the numerical section?

---

> ### Author Response · Authors · 2024-11-22
>
> Dear Reviewer 64fQ,
>
> We thank you for the detailed feedback, we much appreciate it. We address all your concerns below:
>
> Q 1) *I find the paper very confusing and a large time is spent in reviewing background material (e.g. on orthogonal polynomials). It is not clear why one cannot simply use a standard Gaussian quadrature (if the solution is smooth) or a deep Galerkin method with a variational formulation based on finite elements (if the solution is not smooth) and why one needs to learn the weight function.*
>
> Thank you for this question because it helps us emphasize the key positioning of this work. The need of learning weight functions (or some other mechanism offering problem specific adaptivity) is due to a few reasons:
>
> First, the reviewer will likely agree that deep Galerkin methods with finite elements need significant problem-specific engineering. For instance, a key contribution in hp-VPINNs is a careful domain partitioning and test function selection for each problem class (Section 2 in that paper). We eliminate this manual specification: the weight function learns quadrature rules adapted to the PDE structure.
>
> Second, and maybe more importantly, our construction allows solving families of PDEs through a unified scheme. Instead of requiring separate quadrature rules or domain decompositions for each instance, we can now learn the underlying structure shared across the family.
>
> The background on orthogonal polynomials is needed precisely because it helps us show how we make these benefits possible. The connection between weight functions and orthogonal polynomials gives the theoretical foundation for learning quadrature rules that (a) adapt to solution features without intervention (b) while allowing efficient computation through asymptotic expansions.
>
> This is fundamentally different from standard approaches where quadrature rules or test functions are fixed a priori. Further, our framework can be used to solve PDEs in the strong form as well as the weak form.
>
> Q 2) *The experiments seem promising but it is not clear how the L^2 error is computed: is it using the learnt weight function or on a uniform grid. In any case, this requires some convergence analysis to analyze the convergence rate of the proposed quadrature rule as the number of sampling point increases. Otherwise, I do not know why I should use this over any standard quadrature rules.*
>
> Thanks for the question. We compute the $L^2$ test error using $L_2 error= \frac{|| u_\theta - u || }{|| u  ||}$ which we mention in Eqn. 42 in the appendix. We will add a forward reference to this in the main text. Here, $u_\theta$ is the learned solution function and $u$ is the “ground truth” solution. In a small number of cases where the true solution is available in a closed form, we use that as $u$ or alternatively, we use $u$ from a numerical solution obtained using a traditional numerical scheme (finite difference). In any case, the test error is evaluated on a uniform grid of a much higher density (10x) than the number of points used in the training setting. We should clarify that the “ground truth” solution is not used in any form during training. We have included results for convergence on increasing the number of sampling points for our proposed method and other baselines.
>
> We believe that the reviewer would agree that adaptive methods offer various benefits compared to fixed grid techniques. In the context of PINNs a fixed grid would lead to overfitting with poor generalization performance. Several papers \cite{RAR, RAD, R3} study the effectiveness of adaptive sampling for PINNs and reveal that both fixed grid and sampling uniformly from the domain are not the best strategy. As extensively demonstrated in our experiments (Table 1) LearnQuad facilitates learning an adaptive quadrature rule is better than other adaptive baseline methods on a plethora of problems relevant in the literature.
>
> Q 3) *Line 49: I don't understand this paragraph: integrating the equation would just give you an equation between two real number, the first one would correspond to the average of u over the domain. After integration, it is impossible to recover u. To solve (1), one can either solve it in the strong form, or in the weak form after integrating by a test function (smooth with compact support)*
>
> We apologize for the oversight. Thank you. We meant to indicate the weak form integration with a test function (smooth with compact support) and have adjusted the equation and the surrounding text. We meant :
>
> $$
> \int \int \left( \frac{\mathrm{\partial}^2 u}{\mathrm{\partial}x^2}  + \frac{\mathrm{\partial}^2 u}{\mathrm{\partial}y^2} \right) v(x,y)  \mathrm{d}x \mathrm{d}y  = \int \int \left( \log(x) \sin(y) +  f(x,y) {y^3} \right) v(x,y) \mathrm{d}x \mathrm{d}y
> $$
>
> where $ v(x,y) $ is the test function.

---

> ### Author Response · Authors · 2024-11-22
>
> Q 4) *The choice of the quadrature rule is basically equivalent to the choice of basis functions to approximate u. If one uses a finite element method, then one can employ a quadrature rule on each mesh element that integrates polynomial exactly up to the FEM degree, hence making zero integration error, and similarly for spectral methods.*
>
> Thanks for the comment. We agree that the choice of quadrature rule is equivalent to the choice of basis functions. Let us reframe one of our contributions in terms of basis expansions common in the FEM literature. Consider a well-known method like hp-vpinn. The solution in many examples there is approximated using a Legendre polynomial basis on a partitioned domain. The choice of this polynomial is nice, but needs at least a few problem-specific decisions: polynomial degree per element, domain partitioning, and importantly, the weight function that induces the family of orthogonal polynomials. We provide a mechanism to learn some of these choices.
>
> We know that each known orthogonal polynomial is associated with a weight function and a corresponding quadrature rule (pertaining to the roots of the orthogonal polynomial). For example:  Gauss-Legendre quadrature rule is associated with the Legendre polynomial with weight function $w(x)=1$; Gauss-Hermite quadrature rule is associated with the Hermite polynomial with weight function $w(x)=\exp^{-x^2}$; etc. We exploit the orthogonal polynomials stemming from the modified Gauss-Jacobi weight function $w(x)= (1−x)^\alpha(1+x)^\beta h(x)$ where $h(x)$ is a non-negative function. This choice is based in large part on very recent advances in asymptotic expansions of the quadrature nodes and weights of the modified Gauss-Jacobi type orthogonal polynomial.
>
> Hence, instead of fixing the weight function (as in Legendre), we parameterize the modifier $h(x)$ and learn an adapted/modified weight function that induces a problem-adapted family of orthogonal polynomials. This learned family then provides both the quadrature points, or equivalently, an adapted basis for representing the solution. This adaptation happens automatically via optimization rather than the a-priori choice of appropriate basis, and would be difficult to obtain without the asymptotic expansion results we leverage here.
>
> Q 5) *line 98: I would refrain from making such novelty statement*
>
> We are happy to tone down the language and will rephrase this in the updated version.
>
> Q 6) *I do not understand why introducing Cauchy residue theorem is necessary or Section 3, and 5.1.*
>
> We believe that our short explanation below will clarify why these sections are necessary, which serve to bridge a theoretical/conceptual formalism with a practically implementable module.
>
> Consider Section 3: Section 3 provides the foundation by introducing both strong and weak formulations of PDEs. This is not just background: it helps us see how the learnable quadrature approach can serve two roles: we can get collocation points for the strong form, while the weight function can serve as a test function for the weak form. We check both empirically later, and without Section 3, the empirical results will seem awkward without much background context.
>
> Consider Section 5.1: The problem of computing quadrature nodes and weights leads us to asymptotic expansions. The reviewer will agree that direct computation of orthogonal polynomial roots, especially for high degrees, is challenging. The asymptotic expansions provide very stable formulas for these computations. But these expressions involve contour integrals that are difficult to compute within a computational setting involving back-propagation. We believed that most readers will appreciate a clear identification of these technical hurdles.
>
> Consider Cauchy Residue Theorem: The above challenge is precisely where the Cauchy Residue Theorem is crucial. The asymptotic expansions for quadrature nodes involve contour integrals which are correct/elegant, but challenging to differentiate through. By assuming simple poles and applying the Residue Theorem, we can transform these contour integrals into direct function evaluations. This transformation is nice for simple poles and enables the use of automatic differentiation. Without these ideas, implementing the asymptotic expansions within a learnable module would be intractable.
>
> Together, these sections form an essential chain of mathematical development that allows practical implementation. Each component - from the weak/strong formulations to the residue theorem to the asymptotic expansions represents a carefully chosen step in making the theory compatible with gradient-based learning frameworks. Many of these concepts are not common in machine learning, and omitting them would make the formulation quite hard to digest for most readers. We believe these sections make the paper accessible to a much larger audience within our community.

---

> ### Author Response · Authors · 2024-11-22
>
> Q 7) *line 154: this requires the function spaces for u and f. If f=0, then the solution is trivial as u=0, with the choice of boundary conditions.*
>
> Yes, this is correct. Our goal here was pedagogical: simply to introduce the key components (operator L, forcing term f, boundary conditions etc) that appear throughout the paper. We chose this form because many readers are likely to recognize the Poisson equation. The subsequent sections demonstrate our method’s capabilities in more interesting cases (in Section 6). If the reviewer suggests, we are happy to replace it with a more non-trivial example or modify the text to note that the solution is trivial but we are using it for illustrative purposes.
>
> Q 8) *Is there a connection with adaptive mesh refinement?*
>
> While our method shares the broad goal of adaptation with classical adaptive mesh refinement (AMR), there are differences. Traditional AMR methods rely on a posteriori error estimators (either residual-based or recovery-based), fixed refinement criteria, and explicit mesh modification procedures. These require careful problem-dependent choices and often involve solving local problems for error estimation. Our framework is different due to its weight function parameterization. Rather than using predetermined error indicators, the distribution of quadrature points adapts automatically through optimization of h_θ(x) and requires no explicit error estimation.
>
> Q 9) *line 203: the choice of this particular weight function is not motivated*
>
> As noted in our answer to Question 4, our goal of learning the weight function corresponding to a set of orthogonal polynomials is to eventually enable a learnable (or adaptive) quadrature. In order for the method to be practical, we would want to compute these efficiently. Our method exploits asymptotic expansions of quadrature nodes for efficiency. As mentioned in line 266-267 these are recent developments and are most complete for modified Gauss-Jacobi type weight functions, which motivates our choice. Thank you for pointing this out. We agree that mentioning this earlier (around line 203) would be beneficial and have adjusted the text accordingly.
>
> Q 10) *line 215: fix formatting for the equation*
>
> We have fixed the formatting.
>
> Q 11) *line 244: "enable learning", I do not understand what this means*
>
> Our comment regarding standard quadrature was to point out that use of standard quadrature does not allow data-dependent sampling and hence is not suitable for use in a neural network to solve either a single PDE or a family of PDEs. We will rephrase the statement to clarify this further.
>
> Q 12) *line 248: a general statement like this is meaningless as certain equations are very easy to solve. Training a neural network to solve a PDE is often (always?) more computational intensive.*
>
> We agree. The statement around line 248 was indeed too general, and we will revise it to be more precise. While certain PDEs admit easy solutions, the challenges we address are specifically relevant in the following contexts.
> First, for traditional solvers, increasing resolution requires solving progressively larger systems of equations. A neural network approach, once trained, provides a continuous solution. Second, traditional solvers will solve each instance independently, our approach learns the underlying structure and is useful when solving multiple related instances. For one instance, the cost can indeed be higher, but this is amortized when solving multiple instances. Third, parallel calculation of quadrature points can allow benefits in scaling to higher dimensions where we showed preliminary results as a first step.
>
> Q 13) *How do you compute the L2 error in the numerical section?*
>
> The reported test errors are the relative L2 error computed using: $L_2 error= \frac{|| u_\theta - u || }{|| u  ||}$ which we mention in Eqn. 42 in the appendix. We will add a reference to this in the main text. Here, $u_\theta$ is the learned solution function and $u$ is the “ground truth” solution. In a small number of cases where the true solution is available in a closed form we use that as $u$ or we use $u$ to be a numerical solution achieved using a traditional numerical scheme (finite difference). In any case, the test error is evaluated on a uniform grid of a much higher density (10x) than the number of points used in the training scenario. We restate that the “ground truth” solution is not used in any form during the training period.
>
> We sincerely hope that we have been able to address your concerns and provide more clarity. We are very happy to answer any questions you may have. Thanks again for the time and effort in reading our work and providing feedback to improve the paper.

---

> > ### Comment · Reviewer_64fQ · 2024-11-27
> > **Response to authors**
> >
> > I thank the authors for the detailed response and I also read the reply to other reviews. However, I do not think that the authors address my main concern related to Q.1. Since the method proposed by the authors consist of learning a modification of the Gauss-Jacobi weight function, it is not clear to me whether it brings any advantage over just using the Gauss-Jacobi weight function (i.e. does it learn anything?). Secondly, this procedure might be advantageous over PINNs in settings where the solution is smooth (i.e. can be well approximated by polynomials) on a regular, structured domain, which is the area where one would use a spectral method instead to achieve fast convergence. It is not clear to me, based on the authors' experiments, whether there is any hope of improving upon PINNs in a regime where they might be useful (such as on a complicated domain where meshing by be difficult, or high dimensional PDEs). Here, I actually share the same concerns as comments 1 and 2 in the response by Reviewer ante.
> >
> > Finally, I do not find the choice of the weight function to be well-motivated, if one were to learn it, why trying to imposing a partial formulation? Then, one would also want to do some analysis for specific PDEs and show convergence rates for the corresponding learned quadrature.
> >
> > For these reasons, I would like to maintain my original evaluation of the paper.

---

> ### Author Response · Authors · 2024-11-28
>
> Dear Reviewer 64fQ,
>
> We thank you for the time you have invested in our paper and are happy to provide clarifications.
>
> Q) *Since the method proposed by the authors consist of learning a modification of the Gauss-Jacobi weight function, it is not clear to me whether it brings any advantage over just using the Gauss-Jacobi weight function (i.e. does it learn anything?).*
>
> We thank you for the time you have invested in our paper and are happy to provide clarifications. We conducted an experiment to validate the learning of the weight function. For the convection equation we turned off learning the weight function and achieved a $L_2$ relative error of $2.3$ compared to $0.78$ when learning the weight function. This is the experimental setup from Table 1 (column 1) in the current version of the paper. Also, as indicated by the other baselines in the extensive experiments conducted both in Table 1 and Table 2, it is evident that adaptive sampling schemes are indeed beneficial for PINNs.
>
> Q) *Secondly, this procedure might be advantageous over PINNs in settings where the solution is smooth (i.e. can be well approximated by polynomials) on a regular, structured domain, which is the area where one would use a spectral method instead to achieve fast convergence. It is not clear to me, based on the authors' experiments, whether there is any hope of improving upon PINNs in a regime where they might be useful (such as on a complicated domain where meshing by be difficult, or high dimensional PDEs).*
>
> Regarding the choice of problems on which we demonstrated the utility of LearnQuad, we want to reiterate the scope of the paper. Our work is focused on improving the performance of PINNs by incorporating an adaptive sampling scheme and hence we have a clearly defined body of baseline papers [1],[2],[3],[4],[5],[6]. We have included our results for almost all possible PDEs handled in this literature to the best of our knowledge. Should there be more settings, already being handled within the scope of this paper, we are happy to include them.
>
> Q) *Finally, I do not find the choice of the weight function to be well-motivated, if one were to learn it, why trying to imposing a partial formulation? Then, one would also want to do some analysis for specific PDEs and show convergence rates for the corresponding learned quadrature.*
>
> Finally, as mentioned in our previous response regarding the choice of the weight function (Q 9 above), we point out that our goal to learn the weight function is to enable an adaptive quadrature scheme. Without an efficient scheme to compute the quadrature node, this would not be fruitful. Hence we exploit the recent developments in asymptotic expansions of quadrature nodes for the modified Gauss-Jacobi weight functions for which the results are most complete. We believe that the reviewer would likely agree that naively learning a weight function (without imposing any structure) would lead us to first identify the associated orthogonal polynomials and there-after compute the roots of these polynomials to be used as quadrature nodes. We don’t believe there is an easy alternative to these within a learnable framework without relying on asymptotic expansions as utilized in our work.
>
> We sincerely hope that we have been able to address your concerns. We are happy to answer further questions.
>
> [1] Daw et al. Mitigating propagation failures in physics-informed neural networks using retain-resample-release (R3) sampling, ICML 2023.
>
> [2] Lu, Lu, et al. "DeepXDE: A deep learning library for solving differential equations." SIAM review 63.1 (2021)
>
> [3] Wu, Chenxi, et al. "A comprehensive study of non-adaptive and residual-based adaptive sampling for physics-informed neural networks." Computer Methods in Applied Mechanics and Engineering 403 (2023)
>
> [4] Krishnapriyan, Aditi, et al. "Characterizing possible failure modes in physics-informed neural networks." Advances in neural information processing systems 34 (2021)
>
> [5] Nabian, Mohammad Amin, Rini Jasmine Gladstone, and Hadi Meidani. "Efficient training of physics‐informed neural networks via importance sampling." Computer‐Aided Civil and Infrastructure Engineering
>
> [6] Wang, Sifan, Shyam Sankaran, and Paris Perdikaris. "Respecting causality is all you need for training physics-informed neural networks." arxiv 22

---

> ### Author Response · Authors · 2024-12-02
>
> Dear Reviewer 64fQ,
>
> We really appreciate the time you've invested in our paper. If there is anything else we can answer or clarify, please let us know. Many thanks.

---

### Official Review · Reviewer_ante · 2024-11-03

**Soundness:** 2
**Presentation:** 2
**Contribution:** 2
**Rating:** 3
**Confidence:** 4

**Summary:**

The authors discuss a data-driven method to compute quadrature weights for numerical integration used for the solution of partial differential equations. They introduce an algorithm to compute the weight functions, then obtain the quadrature points from it, and solve the PDE, all in an "end-to-end" pipeline. The procedure is demonstrated in several computational examples, solving the heat, wave, Burgers', diffusion, and advection PDEs.

**Strengths:**

The authors address a very important point in the solution of PDEs: where in space to evaluate the operators, and how to construct test functions for weak forms. Numerical integration in general (even beyond PDEs) is also an incredibly important topic, e.g. in the quantification of uncertainties (i.e., expectations that usually involves integrals).
The authors provide a good overview about classical schemes for integral weight constructions at the beginning. Data-dependent choice of test functions for weak forms, and choice of collocation points for strong form solutions, is at the heart of classical solution methods for PDEs, and both topics are being addressed in the paper. Learnable quadrature rule to compute the integral associated with the weak form is equally important, especially on more complicated domains and in higher dimensions, and the method may hold promise to address some of these challenges.

**Weaknesses:**

The structure of the paper, as well as language and typesetting, should be improved significantly. I detail my concerns below. Beyond these issues about first impressions, the numerical experiments that should show the strength of the approach are lacking in detail - even when considering the appendix - and are impossible to reproduce with the given text. One of the main concerns I have with the experiments is that the authors only rely on the PINN loss approach to solving very simple and low-dimensional PDEs, where much better classical methods exist. No comparison to classical approaches is presented, in fact, it seems the "classical solution" is sometimes used as the ground truth, making it very hard to argue why this new method is worth pursuing (if the classical method can outperform it already). Here are some of the concerns in more detail:

 * The structure of the paper is very confusing, and the text does not help to mitigate this. Examples:
   - section 5.2 has a lot of "subsections" with boldface text. This list has multiple different abstraction levels, and does not just consist of "several assumptions or simplifications" as stated at the beginning: "Simple Poles" is completely different from "Root finding and Implicit Function Theorem", which again is different (on an abstract level) from "How to parameterize?".

   - l. 343: "Our goal is to report a solution function u." This sentence is stated in the middle of a section on finding weights / roots of polynomials. The entire papers goal is the latter, not "to report a solution".

 * There are a lot of typesetting and language mistakes. I will not list all, just examples:
   - l. 345: "We use a simple (MLP) based neural network" -> should be "We use a simple MLP-based neural network" (even though it rather should be "We use a MLP")

 * The figures are not designed well and confusing. For example:
   - figure 3: Why is there text "interlacing of roots!" inside the figure? What do the p_n and p_{n+1} stand for, the points or the polynomials? Do the blue points also count for "p_{n+1}"? This is all not clear from looking at the figure, it is necessary to read the entire text (even more than the caption) to understand what is shown.
   - figure 4: the axes labels are way too small.

 * Algorithm 1 is not self-contained either. The "solution function u_\theta" is not referenced, but described in an earlier section "How to parameterize?" - and even there it is not clear how many neurons, how many layers, which activation function etc. is used for u. Also, "Gradient based update" in Algorihm 1 is unclear - is mini-batching used, or is it a non-stochastic algorithm (in which case I doubt in can converge to a reasonably good minimum). Also, why is "theta" a "module", not a set of parameter?


 * l429: "This illustrates the viability of LearnQuad for high dimensional PDEs" -> in this example, the authors try to obtain the solution u(x)=sum_i x_i^2 in 100 dimensions. There are no details given on (a) how many data points were used, (b) how long the computations took, (c) how the test errors were computed, etc. One of the (many) challenges of solving PDEs in very high dimensions is the computation of integrals - which is necessary for the computation of "L2 relative error". How did the authors compute the integral in 100 dimensions? I tried to compare the value of the true u(x) to a simple average over the boundary of the cube (0,1) in 100 dimensions on 100,000 points (which is by far not nearly enough, even for Monte Carlo), and I also obtained a "relative L2 error" of 0.089. So even a constant approximation will yield errors similar to the one in the paper, at least when a (very inaccurate) test error computation is used.

**Questions:**

* Why do the authors use a PINN loss and not a proper PDE solver? The quadrature weights should be amenable for classical solvers, especially in low dimensions, and the accuracy for the obtained PDE solution should be many orders of magnitude better.
 * Table 2: "We report the absolute relative error compared to the numerical solution obtained using the same number of domain points". I do not understand this sentence at all. If a numerical scheme with *the same number of grid points* can be used as a "ground truth" solution, why is the new algorithm necessary at all? Could we not just use the "numerical solution obtained using the same number of domain points" to begin with?
 * For the example in 100 dimensions, how was the L2 error computed?

---

> ### Author Response · Authors · 2024-11-22
>
> Dear Reviewer ante,
>
> We thank you for the detailed feedback. Before we address each concern in detail, we would like to clarify the scope of the paper. Our proposed method is in essence a data-driven approach to sample points from any domain. It shouldn’t be viewed as an alternative to traditional numerical methods or Physics Informed Neural Networks (PINN)s. In fact, the right positioning is as an aid to improve the performance of PINNs as studied in R3[1], RAR[2], RAD[3].
>
> We regret if this was not very clear from the presentation, based on the reviewers comments on the presentation, we have adjusted the text to make this clearer.
>
> We now address the concerns below:
>
> Q 1) *The structure of the paper, as well as language and typesetting, should be improved significantly.*
>
> We are happy to make these changes in the final version.
>
> Q 2) *Beyond these issues about first impressions, the numerical experiments that should show the strength of the approach are lacking in detail - even when considering the appendix - and are impossible to reproduce with the given text.*
>
> The code with extensive documentation will be publicly available. And yes, we will include more details in the appendix as well as on the GitHub repository.
>
>
> [1] Daw et al. Mitigating propagation failures in physics-informed neural networks using retain-resample-release (R3) sampling, ICML 2023.
>
> [2] Lu, Lu, et al. "DeepXDE: A deep learning library for solving differential equations." SIAM review 63.1 (2021)
>
> [3] Wu, Chenxi, et al. "A comprehensive study of non-adaptive and residual-based adaptive sampling for physics-informed neural networks." Computer Methods in Applied Mechanics and Engineering 403 (2023)

---

> ### Author Response · Authors · 2024-11-22
>
> Q 3) *One of the main concerns I have with the experiments is that the authors only rely on the PINN loss approach to solving very simple and low-dimensional PDEs, where much better classical methods exist. No comparison to classical approaches is presented, in fact, it seems the "classical solution" is sometimes used as the ground truth, making it very hard to argue why this new method is worth pursuing (if the classical method can outperform it already).*
>
> This comment relates to a few key points of confusion which we will clarify completely.
>
> **Much better classical methods exist:**  We should clarify that our proposed method is not designed to compete with classical solvers. While for simple problems, classical methods are indeed effective, one motivation behind the sizable PINN literature is an alternative which is advantageous in many scenarios: (a) PINN based solutions are mesh-independent; (b) they rely on automatic-differentiation which are easier to implement; and (c) can handle non-linearity effectively given the universal function approximation properties of neural networks.
>
> So, can we improve the performance of PINN-type methods using an adaptive/learnable and efficient sampling scheme? This is the intended scope of this work.
>
> We can verify that PINN based methods never use “classical solution” as the ground truth in the training procedure at all. It is only used to evaluate a test time performance metric. This is needed in cases where the PDE solution is not given in a closed form, which is true for most scenarios. Because otherwise, the main benefit behind the entire body of work around PINNs becomes unclear. We hope this clarifies the concern regarding not comparing with classical methods.
>
> **Choice of problems/very simple and low-dimensional PDEs:** We present an extensive suite of experiments based on a variety of problems handled in the relevant literature R3[1], RAR[2], RAD[3]. Moreover, as suggested by reviewer 5qE3, we also included 5 additional settings from a recent baseline paper R3[1] in which case our method can outperform existing approaches in 4 out of 5 cases (please see the new results above). The development of PINNs is still relatively young compared to classical numerical analysis, and the problems described in our paper (and other PINN focused papers such as R3[1])  have emerged as good testbed to assess methodological advances in capabilities for physics informed neural network models.
>
> We further explain our choice of PINN loss over PDE solvers in the response to Q10 below.
>
> Our main contribution is not just solving PDEs, but learning how to optimally sample points based on the PDE's structure. We emphasize that advantages stem directly from our core theoretical contribution: the learnable weight function that induces problem-specific quadrature rules. This is fundamentally different from both classical adaptive methods and other existing ML approaches like R3[1], RAR[2], RAD[3]; all of whom invariably rely on computing error estimates through residual-based estimators or gradient thresholding which are problem-specific, need to be chosen carefully, and sometimes may need to solve additional local problems. Instead we adaptively learn where refinement may be needed in an end-to-end fashion in conjunction with the PINN loss and no additional explicit error estimation is required. As shown by extensive experimental results in Table 1 (page9), this leads to rather large  improvements in error compared to other baseline adaptive methods in the PINN literature.
>
> Q 4) *section 5.2 has a lot of "subsections" with boldface text. This list has multiple different abstraction levels, and does not just consist of "several assumptions or simplifications" as stated at the beginning: "Simple Poles" is completely different from "Root finding and Implicit Function Theorem", which again is different (on an abstract level) from "How to parameterize?".*
>
> The reviewer is right that the subsections operate at different levels of abstraction. We chose this structure intentionally, but take the criticism constructively. Here was the logical progression we tried to convey:
>
> (a) Mathematical simplifications: simplification of the contour integral for practical implementation (b) Root finding and implicit function theorem shows how to maintain differentiability, an essential bridge between the formulation and implementation, and finally (c) the implementation details covering specific design choices (how to parameterize), their validation and practical stabilization tricks. We hope this clarifies our reasoning but we appreciate the comment, and will use it to more clearly emphasize the progression from theory to practice very transparent. We have adjusted line 321 to include implementation, which is also mentioned in the subsection heading.

---

> ### Author Response · Authors · 2024-11-22
>
> Q 5) *l. 343: "Our goal is to report a solution function u." This sentence is stated in the middle of a section on finding weights / roots of polynomials. The entire papers goal is the latter, not "to report a solution"*
>
> We are sorry for an incorrect choice of word. We wanted to say that the eventual result will be a solution function, u. We have updated the text accordingly.
>
> Q 6) *There are a lot of typesetting and language mistakes. I will not list all, just examples: l. 345: "We use a simple (MLP) based neural network" -> should be "We use a simple MLP-based neural network" (even though it rather should be "We use a MLP")*
>
> Thank you for pointing this out, we have fixed these.
>
> Q 7) *The figures are not designed well and confusing. For example:figure 3: Why is there text "interlacing of roots!" inside the figure? What do the p_n and p_{n+1} stand for, the points or the polynomials? Do the blue points also count for "p_{n+1}"? This is all not clear from looking at the figure, it is necessary to read the entire text (even more than the caption) to understand what is shown. figure 4: the axes labels are way too small.*
>
> We take this suggestion constructively.
>
> Figure 3: We will remove the text from inside the figure. $p_n$ and $p_{n+1}$ stand for the polynomials of degree $n$ and $n+1$ respectively. The blue points correspond to the roots of $p_{n}$ and red are for roots of $p_{n+1}$. We agree that the caption in this case should be more detailed and we have updated it.
>
> Figure 4: We have increased the axis labels, and adjusted the placement of figures.
>
> Q 8) *Algorithm 1 is not self-contained either. The "solution function u_\theta" is not referenced, but described in an earlier section "How to parameterize?" - and even there it is not clear how many neurons, how many layers, which activation function etc. is used for u. Also, "Gradient based update" in Algorithm 1 is unclear - is mini-batching used, or is it a non-stochastic algorithm (in which case I doubt in can converge to a reasonably good minimum). Also, why is "theta" a "module", not a set of parameter?*
>
> We have updated Alg. 1 to reference $u_\theta$ in place of $\theta$. The details of neural networks (layers, hidden dimension and activation function) have been added to the appendix. The gradient update is stochastic due to varying the degree of polynomial. Yes, by a learnable module, we meant a set of parameters, we have updated the text.
>
> Q 9) *l429: "This illustrates the viability of LearnQuad for high dimensional PDEs" -> in this example, the authors try to obtain the solution u(x)=sum_i x_i^2 in 100 dimensions. There are no details given on (a) how many data points were used, (b) how long the computations took, (c) how the test errors were computed, etc. One of the (many) challenges of solving PDEs in very high dimensions is the computation of integrals - which is necessary for the computation of "L2 relative error". How did the authors compute the integral in 100 dimensions? I tried to compare the value of the true u(x) to a simple average over the boundary of the cube (0,1) in 100 dimensions on 100,000 points (which is by far not nearly enough, even for Monte Carlo), and I also obtained a "relative L2 error" of 0.089. So even a constant approximation will yield errors similar to the one in the paper, at least when a (very inaccurate) test error computation is used.*
>
> This was a feasibility experiment for checking the scalability of the method to higher dimensions and not a main highlight of the paper. We are happy to remove it if it is confusing.
>
> We have included more details in the paper and include them here for completeness. We used 1000 points in 100 dimensions. For this problem, our training took 18 seconds to converge in 300 epochs. After this, evaluating the trained model on any given resolution takes 0.0065 seconds. The test errors were computed with respect to the true analytical solution which is readily available in this case. We should note that we are using PINNs which do not require direct evaluation of the integral and hence offer some advantages including readily generating a continuous solution to the PDE. The “L2 relative error” calculation is in Eqn (42). We note that this is the same setup used in the DeepRitz paper [4] which deploys a Monte Carlo scheme to sample points. As we acknowledged in the paper, this is a simple problem with a smooth solution, hence any learnable sampling scheme should not be expected to provide much performance gains.
>
> [4] Yu, Bing. "The deep Ritz method: a deep learning-based numerical algorithm for solving variational problems." Communications in Mathematics and Statistics 6.1 (2018)

---

> ### Author Response · Authors · 2024-11-22
>
> Q 10) *Why do the authors use a PINN loss and not a proper PDE solver? The quadrature weights should be amenable for classical solvers, especially in low dimensions, and the accuracy for the obtained PDE solution should be many orders of magnitude better.*
>
> Thank you for the question. We believe this can be fully clarified.
>
> First, let us accept that if one is interested in solving a single PDE at a fixed resolution corresponding to a simple problem in lower dimensions, classical solvers are an excellent choice. In this paper, the object of interest is PINN, which we describe next. As mentioned in our response to Q3 above, PINNs provide a mesh independent solution, are more amenable to non-linearities and are easier to scale and implement. Hence, PINNs offer many benefits in several cases and for this reason, are being studied extensively. Next, let us explain why we choose to use a PINN loss instead of a PDE solver.
>
> While we agree with the reviewer that the learnable quadrature rule is amenable for classical solvers, there are several issues. Suppose we use a classical solver instead of a PINN loss. This means that for each update of parameters $\theta$ in $w_\theta$ (which induces the quadrature) we will need to (i) generate quadrature points using current $w_\theta$, (ii) solve the system of equations (either implicitly using a solver or iteratively), (iii) compute some loss/quality and (iv) update \theta to improve this metric. This poses several challenges. Explicit (iterative) solvers are memory-intensive when unrolling across time steps, sensitive to numerical instabilities, thereby requiring fine time steps and increased computational cost. When differentiating through a numerically unstable solver, the gradients can become inaccurate or blow up. Implicit solvers demand solving linear or nonlinear systems. Computing Jacobians for implicit differentiation requires significant computational resources. Furthermore, matrix inversion or solving linear systems as part of implicit differentiation introduces high computational overhead. Hence, we can agree that integrating PDE solvers into neural network modules presents challenges for both explicit and implicit solvers due to the above mentioned issues in computing gradients which are necessary to update the models via back-propagation. Therefore, in order to make learnable quadrature feasible – the main goal of this work – we leverage the PINN loss which is more suited for the end-to-end learning framework.
>
> Another aspect worth mentioning is regarding the setup for a family of PDEs. Without a scheme to learn the common structure shared between different instances of the PDE, it would require solving each instance separately at each desired resolution. To conclude, the choice of the loss is important not for solving individual PDEs, but for permitting the learning of quadrature rules that can then be used across multiple problems/solution schemes.
>
> Q 11) *Table 2: "We report the absolute relative error compared to the numerical solution obtained using the same number of domain points". I do not understand this sentence at all. If a numerical scheme with the same number of grid points can be used as a "ground truth" solution, why is the new algorithm necessary at all? Could we not just use the "numerical solution obtained using the same number of domain points" to begin with?*
>
> We are sorry if the language is unclear, we will rephrase. We compute the test error with respect to a numerical scheme on a fixed uniform grid. We clarify that this information is not used during training. In this setting where our model is learning for a related set of PDEs, the model is trained only once and then we can generate solutions to any PDE from the set at any desired resolution with a single forward pass, without the need to do any fine-tuning at all. This, when compared to using a numerical solution, has two advantages. First, we do not need to solve the PDE for every new PDE instance and second, we do not have to re-solve for any new resolution at which we want to recover the solution. These are some benefits offered by a PINN based solution.
>
> Q 12) *For the example in 100 dimensions, how was the L2 error computed?*
>
>  In this case, the true solution has a closed form and we use it to compute the test error. We note that this is not used during the training procedure. We have mentioned the formula to compute “L2 relative error” in Eqn(42) and will refer to it in the main text. We note that this is the same setup used in the DeepRitz paper [4] which uses a Monte Carlo scheme to sample points.
>
> We hope that we have been able to address the concerns raised by the reviewer. We are very happy to provide further clarification at any point, and then upload the revised paper with all the changes.

---

> > ### Comment · Reviewer_ante · 2024-11-26
> > **Paper modifications and more challenging problems are missing**
> >
> > I appreciate the detailed answers to my comments, and the new experiments reported on in the comments to other reviewers. However, the key issues I raised in my initial feedback are still not addressed. I hope I can make them more precise below.
> >
> > [1] The PDF is not updated (as far as I can see?), so I cannot judge how the new figures look like and how the text is improved. The authors claim they changed this in a new version, so I urge them to upload it soon.
> >
> > [2] Regarding "PINN vs. classical solver": I appreciate the detailed answers and justification regarding "learning good collocation points for PINNs". It is a valid direction to try to improve PINNs (there is a lot that must be improved), and I also now understand why the PINN loss is necessary and not just added for convenience. However, the current comparisons are far away from what the authors claim PINNs are able to do. The numerical experiments in the current manuscript are for very low-dimensional cases (1D only), with parameter spaces that are equally low-dimensional. The computational effort for training the models and quadrature for these examples is completely disproportional to how much effort it would take to solve them classically (much faster, more accurate), so they do not demonstrate the benefits the authors claim that PINNs have. It is not reasonable to argue that "with classical methods, the PDEs have to be solved again each time" - in fact, one could use classical solvers to solve the (paramerized!) PDEs 100 times very quickly and accurately over the entire parameter space, and then just train a neural network in a supervised way (or interpolate the generated data in some other way). This would be important to demonstrate that the new approach has any benefits for the given examples. If the authors claim that they want to improve PINNs because they have favorable properties compared to some classical solvers, they should demonstrate them on examples that show this. I understand that there is no time to do this in the short review time period of this conference, but in the current paper I cannot assess if learning quadrature points has any benefit for PINNs that help them *in settings where they are actually useful*. The last sentence is very important for my criticism: comparing PINN approaches in settings where PINNs are outperformed by classical methods and then claiming that PINNs are better in some other settings does not help to assess if the method helps in those other cases.
> >
> > [3] Regarding the high-dimensional example: the authors do not replicate exactly the experiment in the "Deep Ritz" paper. The authors of DeepRitz use 1000 points in *each* iteration (of which they use 50,000 for the d=100 case), with additional 100 points *per hyperplane* on the boundary. Even in that case, the result is only slightly more accurate than my simple experiment of comparing to the average of the values on the boundary. Also, my main criticism is not about the number of training data points, it is purely on the number of points used *to compute the test error*: this error is defined as an integral ($L^2$ error between functions), so practically, a 100-dimensional integral must be approximated each time a test error is computed. I apologize that this was not clear, I was not asking about the formula for (relative) $L^2$ error or about the analytic form of the true solution, my concern was how the integral is actually computed. I indeed think the example shows any beneficial property and should be removed - but this also leaves only the low dimensional examples (see [2]). For $d=5$ or even $d=10$ I can see how a carefully chosen integration method can be used to accurately estimate the $L^2$ error, but not for $d=100$.

---

> ### Author Response · Authors · 2024-11-28
>
> Dear Reviewer ante,
>
> We thank you for the time you have invested in our paper. We clarify your concerns in detail:
>
> [1] We have now uploaded the revised version of the pdf.
>
> [2] We are happy that we were able to convey why we chose the PINN loss. We believe that the reviewer will agree that the use of LearnQuad improves the performance of PINNs in almost all settings over both adaptive and non-adaptive sampling schemes as demonstrated extensively in our results (Table 1, 2). The proposed adaptive sampling improving the performance of PINNs is really the core contribution and novelty of our work. As noted in previous comments, we have covered all prominent examples of PDEs from the PINN literature. While some works around PINNs might come across, based on how those papers are positioned, as an alternative to classical methods, this is not the intended message of our work, but provides a very reasonable and attractive alternative. Even if one were to consider only lower dimensional PDEs, PINNs are still advantageous given their resolution independent solution. Traditional methods inherently produce solutions at specific grid locations, and for any other location we have to resort to either interpolation or as the reviewer suggested, train another neural network.
>
> We need to acknowledge that interpolation between solutions (obtained on discrete grids) is not guaranteed to solve the PDE and the accuracy of the interpolated solution largely depends on the resolution of the grid, the smoothness of the solution as well as the interpolation method.
>
> The training of another neural network in a supervised format, will be more time consuming than training the PINN in one go, even if one ignores the cost of collecting the training data. This is indeed the approach in the area of operator learning, which we have contrasted with in Remark 6.3. Further, the other advantage of PINNs in these settings is the memory footprint. The PINN solution function trained in our experiments always had fewer than 500 parameters. We can compare this to the case where one had to store the solution for the PDE over multiple grid locations (and multiple resolutions), derived from a traditional solver. With the use of PINNs one can effectively store 500 values and generate the solution at any desired resolution.
>
> While training a PINN model is time-consuming, it's worth noting that traditional solvers also took years of development and refinement to achieve their current efficiency. High-quality CFD solvers require substantial budgets for licensing and decades of research, as is the case with weather forecasting. While we don't claim that PINNs will replace classical methods, they offer a promising alternative. This work focuses on enhancing PINNs, recognizing that significant progress still lies ahead.
>
> We see a similar trend in the field of NeRF. The great body of work in NeRF also did not perform better than solvers right away. But over time, research evolved, and the use of implicit neural networks in capturing scenes is now taking over the solver approach of fixed voxel discretization. Similarly, since optimism drives fundamental research, we dedicate this work to improve PINNs.
>
> [3] We clarify the computation of the test error of 100 dimensions below. Firstly, both us (and DeepRitz) compute the error over the discretized domain and hence avoid the computation of the integral. The $L_2$ error is using the discrete norm rather than the continuous one over function spaces. The discretized domain is an uniform grid and hence we have the following formula.
>
> $$ \text{Relative} L_2 \text{error} = \frac{|| u_\theta(x) - u(x) ||_2}{|| u(x) ||_2} $$
>
> where: $$ || u_\theta(x) - u(x) ||_ 2 = \sqrt{\frac{1}{n} \sum_{i=1}^n  ( u_\theta(x_i) - u(x_i) )^2 }$$
>
> and $$ || u(x) ||_ 2 =  \sqrt{\frac{1}{n} \sum_{i=1}^n  ( u(x_i) )^2 $$
>
> where $x_i$ are the points sampled in the 100 dimensional space.
>
> The need for an integral is avoided because the summation over sample points replaces it. We very densely sample a sufficient number of points, (10000, uniformly) to capture the function’s behavior in 100 dimensions.
>
> We sincerely hope that we have been able to address your concerns. We are happy to answer further questions.

---

> ### Author Response · Authors · 2024-12-02
>
> Dear Reviewer ante
>
> We really appreciate the time you've invested in our paper. If there is anything else we can answer or clarify, please let us know. Many thanks.

---

> ### Comment · Reviewer_ante · 2024-12-02
> **Final review**
>
> I appreciate the effort. Unfortunately, none of my concerns have been addressed adequately in the new PDF (still language mistakes, still figures with small axes labels, etc.), so I keep my score as is. I also comment on [W2] and [W3] above again, because it seems the authors did not understand my earlier comments.
>
> [W2] My earlier comment that "the current comparisons are far away from what the authors claim PINNs are able to do" is still valid. It is absolutely valid to try to improve a method even if it is not yet at the state of the art, but the current paper does not even discuss the state of the art, or includes any of the state of the art methods. This means a reader without knowledge of these PDEs would assume that the method is actually state of the art - which is misleading. Claiming that "The training of another neural network in a supervised format, will be more time consuming than training the PINN in one go" is not valid, because I can easily  claim that "training PINNs (with the PDE as regularizer in the loss function) can be much more complicated (in terms of the loss landscale and number of hyperparameters) than training a network using supervised learning". Again, I can bring the same argument: the authors should not write statements without backing them up with experiments.
>
> [W3] "The need for an integral is avoided because the summation over sample points replaces it." I understand that the integral is not computed analytically, but it is certainly approximated using the sum. For a scalar, real-valued function $u$ on a domain $X$, the $L^2$ norm is defined by $|u|_2=L^2(u)=\sqrt{\int_X |u(x)|^2 dx}$.
>
> This integral can be approximated by a Monte-Carlo sum, so that $L^2(u)\approx \sqrt{\frac{1}{N}\sum_{k=1}^N |u(x_k)|^2}$. This seems to be what the authors do (equation 41 in the new manuscript, for the relative $L^2$ loss, and their answer above). Monte-Carlo approximation of integrals has a convergence rate of $\frac{1}{\sqrt{N}}$, which is quite slow, but technically independent of the input dimension. The authors do not demonstrate that 10,000 points are enough to actually reach the regime where this convergence rate is valid - and in $d=100$ dimensions, I strongly doubt this, and my quick estimation with the constant function (as detailed in my very first comment) also confirmed it for me. This number of points would normally not even suffice to accurately approximate the integral in 5-10 dimensions, let alone 100.

---

> ### Author Response · Authors · 2024-12-04
>
> Thank you for your continued engagement.
>
> a) *Regarding language and figure quality*: You mention “language mistakes” and “small axis labels” but provide no specific examples. We have carefully reviewed the paper and all figures – all plots use standard font sizes and have readable axes. We can certainly make it bigger but would appreciate any specific adjustments.
>
> b) *Regarding experimental comparisons and state of the art*: You mention state of the art methods but do not specify which ones are missing in our discussion so far. We have extensively compared against recent baselines including several ideas published in the last 12 months in prominent venues as suggested by Reviewer 5qE3, and to our knowledge, have covered all relevant literature related to adaptive and non-adaptive sampling in PINNs. If specific baselines that have not come up so far in the review or the discussion are clearly identified, we are still happy to include these comparisons.
>
> c) Your suggestion regarding supervised learning describes what we interpret as operator learning, where solutions are learned from data either through simulations, measurements or running solvers at fixed resolutions. This approach differs fundamentally from ours in that we neither solve the PDE multiple times nor do we need to ensure that the sampling covers the solution features. The paper describes a formulation to avoid these issues entirely.
>
> d) *Regarding high-dimensional error calculation*: While your discussion of Monte Carlo convergence rate is correct, it overlooks specifics of our setting. For the 100 dimensional Poisson equation, the solution has a simple form. We have verified the stability. Based on your comment, we also increased the number of points to 500K and even up to 1 million. The results change only at the third or fourth decimal place, confirming that our evaluation protocol is sufficient. Furthermore, we note that the high dimensional example is a feasibility study and not the highlight of the paper.
>
> To summarize, both our method and all baselines use identical error calculation protocols, ensuring that the comparisons are fair. If this is insufficient, we would welcome any specific suggestions or alternative metrics that have been used in the PINN literature. We value the time invested in the critique and for engaging with us – but we notice that much of the discussion has centered on broader questions about PINNs versus classical solvers, rather than our paper's specific contribution - a novel construction for learning quadrature rules that demonstrably improves PINN performance. The core criticisms through most of this discussion period appear to challenge the core premise of physics-informed neural networks rather than addressing our technical innovations in quadrature point selection, which are valuable contributions to the field independent of one's position on PINNs versus classical methods. While the development of PINNs to match classical solver performance is an important long-term goal, our work provides concrete advances in sampling strategy that are valuable regardless of where PINNs ultimately find their niche. We welcome specific suggestions (including references to any specific state of the art approaches that are not included in our work).
>
> Thank you.

---

### Official Review · Reviewer_5qE3 · 2024-11-03

**Soundness:** 3
**Presentation:** 3
**Contribution:** 2
**Rating:** 6
**Confidence:** 3

**Summary:**

This work proposed a method to sample the weights and locations of the collocation points used in physics-informed neural networks. The work proposes a data-driven method for sampling points by leveraging asymptotic expansions. The framework is provided for the strong and weak forms. For the strong form of PDEs weight function is proposed for collocation points, and for the weak form of PDEs, the weight function is used as a test function. The method's performance is also shown for a high-dimensional problem and for solving similar problems through hypernetworks.

**Strengths:**

The paper is nicely written and readable for the community. The paper tackles an existing problem in physics-informed neural networks. Choosing collocation points and their weights is challenging, and several advancements in PINNs have tried to tackle this problem.

Several comparisons are provided with baselines showcasing that the method performs well compared to other problems.

With PINNs, one instance of PDE is generally solved, and with neural operators, a family of problems is solved concurrently. However, the latter requires a pre-computed dataset, often from a numerical solver. This paper tries to bridge the gap (although not the first paper to do so) in solving a family of problems through a PINN-based approach where no labeled datasets are required.

**Weaknesses:**

The number of collocation points taken in the numerical experiments is relatively low. It is not discussed why such a setting is kept for baseline methods, which typically require hundreds and thousands of points.

The presentation of Figures could be improved for better readability for the readers.

The choice of parameters in weight function is chosen empirically. The convergence of the proposed method may be largely dependent on those parameters.

**Questions:**

First of all, most of the considered problems have smooth solutions, which are even tractable by the vanilla methods in the literature. It would be a nice experiment to see how the method performs for high-frequency functions/chaotic or multiscale problems.

The experiment provided for high dimension shows no significant gains with respect to the traditional Monte-Carlo approach. The authors reason the smooth solution to be the reason. Can the author perform a similar experiment for a problem that can help understand whether the method is advantageous in higher dimensions?

The authors have not compared their method with R3 [1]. Can the authors explain/compare the advantages/disadvantages compared to R3, one of the known methods in the adaptive collocation domain?

It would be nice to see the comparisons of the presented method with the baselines for more collocation points. Also, it is mentioned that the memory and computational costs of all the methods are almost identical. Does this relationship remain the same in the case with more collocation points? A quantitive comparison will also aid in assessing the method's training efficiency. In general, why is the method not trained with more collocation points?

Performing experiments for different parameter values of the proposed method would help readers to assess the robustness of the proposed method.

[1] Daw et al. Mitigating propagation failures in physics-informed neural networks using retain-resample-release (R3) sampling, ICML 2023.

---

> ### Author Response · Authors · 2024-11-22
>
> Dear Reviewer 5qE3,
>
> We thank you for the detailed feedback. We appreciate your acknowledgment of the problem the paper proposes to handle and your overall positive view. We address your concerns below:
>
> Q 1) *The number of collocation points taken in the numerical experiments is relatively low. It is not discussed why such a setting is kept for baseline methods, which typically require hundreds and thousands of points.*
>
> We should clarify a couple of points. First, other baseline papers including R3[1], RAR[2], RAD[3] also use a similar number of points in their experiments. While the number of collocation points may seem low, the goal in these other papers and also our work is to check whether we can get a numerically desirable solution without an excessively large number of points. For all these methods, we see that increasing the number of points almost universally improves results. We should note that this is also noted in R3[1], where a small and fixed budget in terms of compute memory and time means that one is limited a smaller number of collocation points to train the neural network based solution function, and is the more challenging setting. To fully answer this question, we have included additional convergence results obtained when increasing the number of points which shows that increasing the number of points will only help LearnQuad converge faster.
>
> Interestingly, the regime of a large number of points is quite favorable to LearnQuad: increasing the number of collocation points, due to the use of asymptotic expansions and simplifications (in section 5), leads to a very efficient solution. In particular, we emphasize that our parameterization of expansion coefficients and circumventing contour integrals (section 5) altogether lead to a parallel computation for quadrature nodes without which these would have a linear time complexity.
>
> Q 2) *The presentation of Figures could be improved for better readability for the readers.*
>
> Thank you for pointing this out, we have updated the captions and Figure 3 and Figure 4 to make it more readable.
>
> Q 3) *The choice of parameters in weight function is chosen empirically. The convergence of the proposed method may be largely dependent on those parameters.*
>
> Yes, the values of $\alpha$ and $\beta$ were chosen empirically but we didn’t observe much variability in performance. Based on this comment, we have now performed a more detailed experiment and will include these plots, which are in agreement, in the revised version of the paper.
>
> Q 4) *First of all, most of the considered problems have smooth solutions, which are even tractable by the vanilla methods in the literature. It would be a nice experiment to see how the method performs for high-frequency functions/chaotic or multiscale problems.*
>
> Our choice of PDEs in the experiment section was guided by prior work. If there is a specific problem that has been omitted but studied in another work, we are very happy to include it.
>
> We are grateful for the suggestion of using R3 as a baseline. Thank you! We now have very promising results for two variants of the convection equation and another new variant for the Allen-Cahn equation described in R3. This suggestion materially improves the experimental section of the paper.
>
> Q 5) *The experiment provided for high dimension shows no significant gains with respect to the traditional Monte-Carlo approach. The authors reason the smooth solution to be the reason. Can the author perform a similar experiment for a problem that can help understand whether the method is advantageous in higher dimensions?*
>
> Our high-dimensional experiment was included to demonstrate scalability, specifically, that the asymptotic expansions and parallel computation of quadrature points remain viable in higher dimensions. While the smooth solution in this case does show comparable performance to Monte Carlo sampling, this experiment serves a different purpose than direct performance comparison. The reviewer will agree that the challenge in the high dimensional setting affects both traditional and learning-based methods. We wanted to check whether our architecture can even work here (such experiments are not shown in most baselines).We acknowledge that realizing this potential for non-smooth high-dimensional problems would require additional developments.
>
> [1] Daw et al. Mitigating propagation failures in physics-informed neural networks using retain-resample-release (R3) sampling, ICML 2023.
>
> [2] Lu, Lu, et al. "DeepXDE: A deep learning library for solving differential equations." SIAM review 63.1 (2021)
>
> [3] Wu, Chenxi, et al. "A comprehensive study of non-adaptive and residual-based adaptive sampling for physics-informed neural networks." Computer Methods in Applied Mechanics and Engineering 403 (2023)

---

> ### Author Response · Authors · 2024-11-22
>
> Q 6) *The authors have not compared their method with R3 [1]. Can the authors explain/compare the advantages/disadvantages compared to R3, one of the known methods in the adaptive collocation domain?*
>
> Excellent suggestion, many thanks. We ran experiments comparing our method with R3 in the settings outlined in the paper (with a 1000 collocation points). In 4 out of 5  cases, LearnQuad has the best performance in terms of relative L2 error (table below). Our performance for Allen-Cahn is comparable. We note that these are initial runs, without an extensive hyper-parameter search for LearnQuad and so, we expect that we can improve LearnQuad’s performance for Allen-Cahn. We will include the paper R3 as a baseline in the revised version of the paper.
>
> For ease of readability, in the updated version, we will briefly point out some formulation distinctions between LearnQuad and R3. R3 samples collocation point(s), always from a uniform distribution, and the adaptive scheme is based on an error estimate similar to traditional mesh refinement schemes. However, LearnQuad samples collocation point(s) from the distribution induced by the learned weight function corresponding to the orthogonal polynomials.
>
>
> | **PDE**          | **Convection (β=30)** | **Convection (β=30)** | **Convection (β=50)** | **Convection (β=50)** | **Allen-Cahn**   |
> |--------------------|-----------------------|-----------------------|-----------------------|-----------------------|------------------|
> | **Epochs**  |     100k                 | 300k                 | 150k                 | 300k                 | 200k            |
> | PINN Fixed                   | 107.5 ± 10.9         | 107.5 ± 10.7         | 108.5 ± 6.38         | 108.7 ± 6.59         | 69.4 ± 4.02      |
> | PINN Dynamic                  | 2.81 ± 1.45          | 1.35 ± 0.59          | 24.2 ± 23.2          | 56.9 ± 9.08          | 0.77 ± 0.06      |
> | Curr Reg                     | 63.2 ± 9.89          | 2.65 ± 1.44          | 48.9 ± 7.44          | 31.5 ± 16.6          | --               |
> | CPINN Fixed                   | 138.8 ± 11.0         | 138.8 ± 11.0         | 106.5 ± 10.5         | 106.5 ± 10.5         | 48.7 ± 19.6      |
> | CPINN Dynamic                | 52.2 ± 43.6          | 23.8 ± 45.1          | 79.0 ± 5.11          | 73.2 ± 3.6           | 1.5 ± 0.75       |
> | RAR-G                        | 10.5 ± 5.67          | 2.66 ± 1.41          | 65.7 ± 1.77          | 43.1 ± 28.9          | 25.1 ± 23.2      |
> | RAD                          | 3.35 ± 2.02          | 1.85 ± 1.90          | 66.0 ± 1.55          | 64.1 ± 11.9          | 0.78 ± 0.05      |
> | RAR-D                        | 67.1 ± 4.28          | 32.0 ± 25.8          | 82.9 ± 5.96          | 75.3 ± 9.58          | 51.6 ± 0.41      |
> | $L^\infty$                  | 66.6 ± 2.35          | 41.2 ± 27.9          | 76.6 ± 1.04          | 75.8 ± 1.01          | 1.65 ± 1.36      |
> | R3                            | 1.51 ± 0.26          | 0.78 ± 0.18          | 1.98 ± 0.72          | 2.28 ± 0.76          | **0.83 ± 0.15**  |
> | Causal R3                    | 2.12 ± 0.67          | 0.75 ± 0.12          | 5.99 ± 5.25          | 2.28 ± 0.76          | **0.71 ± 0.007** |
> | LearnQuad                    | **0.78 ± 0.002**     | **0.68 ± 0.02**      | **0.79 ± 0.02**      | **0.76 ± 0.01**      | 0.87 ± 0.01      |

---

> ### Author Response · Authors · 2024-11-22
>
> Q 7) *It would be nice to see the comparisons of the presented method with the baselines for more collocation points. Also, it is mentioned that the memory and computational costs of all the methods are almost identical. Does this relationship remain the same in the case with more collocation points? A quantitative comparison will also aid in assessing the method's training efficiency. In general, why is the method not trained with more collocation points?*
>
> Yes, we have now included these experiments. Thanks. Yes, the memory and time complexity remain identical with baseline methods even with more collocation points. We will update the appendix with runtime and memory requirements.
>
> Our choice for the number of collocation points was based on those reported in the baselines. One of the more recent works, R3 [1] states “We particularly chose a small value of Nr to study the effect of small sample size on PINN performance” (on page 8 of the paper).
>
> We see the above behavior in our experiments as well, where it is more challenging to learn a neural network based solution for a PDE with a smaller number of points: it may lead to poor generalization. There is another interesting point we want to draw attention to. With more collocation points, the compute requirements of all methods increase as expected. When the number of collocation points are increased, the use of asymptotic expansions within LearnQuad becomes more accurate! Even for a very large number of points, the cost is not prohibitive at all due to the assumptions/design choices in Section 5 which enable efficient computation via parallelization. Thanks again for requesting comparisons with R3.
>
> Q8) *Performing experiments for different parameter values of the proposed method would help readers to assess the robustness of the proposed method.*
>
> Thank you for the suggestion, we will include additional ablation studies for the different parameter values in the final version.
>
> We hope that we have been able to address the concerns. We are very happy to provide further clarification on any point.

---

> > ### Comment · Reviewer_5qE3 · 2024-11-26
> > **Response to authors**
> >
> > Thanks for providing additional experiments. However, I believe the current evaluation of the paper is ideal, and I would stick to my current score.

---

> ### Author Response · Authors · 2024-11-28
>
> Dear Reviewer 5qE3,
>
> We thank you for the positive evaluation of our work. If you have any further questions, please let us know, we are happy to answer them. We sincerely hope that the additional experiments and clarifications can lead to an improvement in the score.
>
> Thank you

---

### Meta-Review · Area_Chair_iWPC · 2024-12-18

**Metareview:**

The manuscript considers a data-driven approach to obtain quadrature weights for numerical integration used for the solution of partial differential equations in the framework of PINN. Overall the approach seems interesting, while the presentation lacks clarity and causes some confusion to the reviewers. While some of those are clarified during the discussion process, overall the reviewers are still not convinced by the approach. After reading the manuscript and the discussions carefully, the meta-reviewer tends to agree that in the current form, the manuscript does not meet the bar of acceptance at ICLR.

**Additional Comments On Reviewer Discussion:**

The authors have tried to address the confusions and comments by the reviewers, however, some of the concerns of the reviewers remain.

---

### Decision · Program_Chairs · 2025-01-22

Reject